# Buoy measurements of strong waves in ice amplitude modulation: a signature of the impact of sea ice closedness on waves in ice attenuation

Jean Rabault[1], Trygve Halsne[2,3], Ana Carrasco[1], Anton Korosov[4], Joey Voermans[5], Patrik Bohlinger[2], Jens Boldingh Debernard[1], Malte Müller[1,9], Øyvind Breivik[2,3], Takehiko Nose[6], Gaute Hope[2], Fabrice Collard[7], Sylvain Herlédan[7], Tsubasa Kodaira[6], Nick Hughes[8], Qin Zhang[8], Kai Haakon Christensen[1,9], Alexander Babanin[5], Lars Willas Dreyer[9], Cyril Palerme[1], Lotfi Aouf[10], Konstantinos Christakos[2], Atle Jensen[9], Johannes Röhrs[1], Aleksey Marchenko[11], Graig Sutherland[12], Trygve Kvåle Løken[13], and Takuji Waseda[6]

[1]Norwegian Meteorological Institute, Oslo, Norway
[2]Norwegian Meteorological Institute, Bergen, Norway
[3]The University of Bergen, Bergen, Norway
[4]Nansen Environmental and Remote Sensing Center, Bergen, Norway
[5]The University of Melbourne, Melbourne, Australia
[6]The University of Tokyo, Kashiwa, Japan
[7]OceanDataLab, Locmaria-Plouzane, France
[8]Norwegian Meteorological Institute, Tromsø, Norway
[9]The University of Oslo, Oslo, Norway
[10]Météo France, CNRM-CNRS, UMR-3589, Toulouse, France
[11]The University Center in Svalbard, Longyearbyen, Norway
[12]Environment Climate Change Canada, Dorval, Canada
[13]Nord University, Bodø, Norway

**Correspondence:** Jean Rabault (jean.rblt@proton.me)

**Abstract.** The Marginal Ice Zone (MIZ) forms a critical transition region between the ocean and sea ice cover, as it protects the close ice further in from the effect of the steepest and most energetic open ocean waves. As waves propagate through the MIZ, they become exponentially attenuated. Unfortunately, the associated attenuation coefficient is difficult to accurately estimate and model, and there are still large uncertainties around which attenuation mechanisms dominate depending on the conditions. This makes it difficult to predict waves in ice attenuation, as well as sea ice breakup and dynamics. Here, we report in situ observations of strongly modulated waves in ice amplitude, with a modulation period of around 12 hours. We show that simple explanations, such as changes in the incoming open water waves or the direct effect of tides and currents and bathymetry on the propagating waves, cannot explain the observed modulation. Therefore, the wave height modulation observed in the ice comes from a modulation of the waves in ice attenuation coefficient. We gather evidence that sea ice convergence and divergence is likely the factor driving this modulation in the attenuation coefficient, through its influence on the ice "closedness". This implies that the level of sea ice "closedness" needs to be taken into account by future dissipation parameterizations.

# 1 Introduction

Sea ice is an important component in the global climate and weather system: in their respective late winters, Arctic sea ice typically covers around 15.5 million square kilometers, and Antarctic sea ice approximately 18.5 million square kilometers (averages are calculated for the reference period 1981-2010, and the trend of the extent of Arctic sea ice in September is decreasing by about 12 percent per decade) (NSIDC, accessed 2024-03; Meier and Stroeve, 2022). Averaged over the year, sea ice covers approximately 25 millions square kilometers (although the trend is decreasing), corresponding to about 7% of the area of the global oceans (Parkinson et al., 1997). Sea ice therefore has a major effect on the energy fluxes and physical oceanography of the polar regions, and a large impact on the Earth's weather and climate dynamics as a whole (Budikova, 2009; Overland et al., 2011; Gao et al., 2015). Moreover, sea ice places drastic constraints on human activities, in particular in the Arctic, where shipping, fishing, logistics, and industrial activities are increasing (Olsen et al., 2020; Shi et al., 2019; Müller et al., 2023).

A particularly important region of sea ice is the Marginal Ice Zone (MIZ). The MIZ is the area of sea ice that is heavily influenced by phenomena happening in the open ocean, in particular, surface waves. The interaction and two-way coupling between the MIZ and the incoming surface waves and swells has a large impact on several key mechanisms of sea ice, such as the breakup, melting and drift of sea ice (Dumont, 2022; Horvat, 2022). This coupling can have large-scale effects on the extent and evolution of sea ice, especially since there are a number of two-way coupling mechanisms (Thomson, 2022; Iwasaki, 2023; Gao et al., 2022). For example, sea ice breakup and melting is currently leading to the appearance of new open water areas in the Arctic, which in turn provides more fetch for waves to grow and break even more sea ice (Thomson and Rogers, 2014). Similarly, albedo and solar radiation absorption mechanisms lead to self-reinforcing dynamics: since sea ice reflects up to 85% of incoming solar radiation, while the ocean absorbs up to around 90% of solar radiation, sea ice melting leads to a self-reinforcing solar radiation energy absorption and temperature increase (Shao and Ke, 2015).

The two-way coupling between waves and ice in the MIZ is related to a number of physical phenomena, in particular:

- Wave attenuation (Squire et al., 1995; Zhao et al., 2015; Zhao and Shen, 2015; Sutherland and Rabault, 2016; Squire, 2020; Løken et al., 2021): As waves propagate in the MIZ, they are progressively attenuated by a number of mechanisms. Although it is not completely clear at present which mechanisms dominate under which conditions, most attenuation mechanisms result, in theory, in an amount of energy dissipation that is proportional to the local wave energy, hence an exponential wave damping as a function of propagation distance. More specifically, the underlying physics can be associated with a combination of viscous damping at the water-ice interface (Zhao et al., 2015; Sutherland et al., 2019; Rabault et al., 2017; Marchenko, 2018), turbulence (Voermans et al., 2019; Smith and Thomson, 2020), viscoelasticity (Mosig et al., 2015; Zhang and Zhao, 2021; Zhao and Shen, 2018; Marchenko et al., 2021a), scattering (Bennetts et al., 2010; Kohout and Meylan, 2008; Montiel et al., 2016; Zhao and Shen, 2016; Bennetts et al., 2024), collisions and floe-floe interactions (Herman et al., 2019; Løken et al., 2022; Herman et al., 2019; Smith and Thomson, 2020).

- Sea ice breakup (Montiel and Squire, 2017; Voermans et al., 2020; Ren et al., 2021; Zhang and Zhao, 2021; Mokus and Montiel, 2022): waves with sufficient amplitude, and applied for long enough, can break the sea ice following a number of flexion and fatigue mechanisms (Voermans et al., 2020; Herman, 2017).

- Sea ice drift and melting (Sutherland et al., 2022): once the waves have broken the sea ice, sea ice can drift more freely, leading to different drift properties. In addition, wave attenuation combined with wave momentum conservation results in gradients of wave radiation stress, which can also be a mechanism that drives the drift of sea ice in the MIZ and the displacement of the ice edge (Dai et al., 2019; Thomson et al., 2021b; Thomson, 2022).

- Fetch modification effects resulting from the presence or absence of sea ice (Thomson and Rogers, 2014; Brenner and Horvat, 2024): sea ice, by modulating the ocean-atmosphere coupling, affects how much effective fetch is available to cause the growth of waves in ice-infested waters.

The combination of these effects creates complex dynamics in the MIZ. This is even more challenging due to the intrinsic complexity of sea ice as a material: depending on temperature, salinity, and the history of the mechanical stress and temperature experienced by sea ice, its mechanical properties can vary by up to 1 order of magnitude, and significant changes could possibly take place even within a timescale of a few days (Williams et al., 2013; Karulina et al., 2019; Voermans et al., 2023). Moreover, additional variables, such as the floe size distribution (FSD) (Wang et al., 2016; Roach et al., 2019; Horvat et al., 2016; Herman et al., 2018), are making the whole picture even more complex, as, e.g., FSD can change quickly through the effect of wind and waves (Wang et al., 2016), and modulate all aspects of sea ice dynamics.

In addition to the intrinsically complex physics involved in sea ice dynamics and the MIZ, sea ice and waves in ice in situ data are relatively scarce. Although several expeditions are conducted on sea ice every year, instrumentation deployments have traditionally been limited due to the intrinsic cost of waves in ice buoys. Multi-buoy deployments have been performed for several decades, e.g. Doble et al. (2006); Thomson et al. (2013); Kohout et al. (2016), and more similar studies. However, recently a new trend has emerged, consisting of the deployment of larger numbers of waves in ice buoys, utilizing open source technologies to reduce the cost of individual buoys by approximately an order of magnitude compared to the commercial instrumentation that has traditionally been available (Rabault et al., 2020, 2022; Kodaira et al., 2023). This approach, which has become more common following the development of low-cost open source instrumentation, e.g., the OpenMetBuoys (OMBs) series of instruments (Rabault et al., 2016, 2020, 2022), and other similar instruments (e.g. the MicroSWIFT of Thomson et al. (2023), and other similar devices described by Cavaleri et al. (2025)), helps to provide larger and more representative amounts of data about the MIZ. The reduced cost of the buoys allows the deployment of up to several tens of buoys during a single expedition (for example, 34 buoys in the course of a single 3-week expedition in Müller et al. (2025)). These larger dataset provide better statistical representations and samplings of the sea ice dynamics. Moreover, deploying large swarms of buoys increases the probability of randomly sampling interesting dynamics and to obtain clusters of buoys even several days or weeks after deployment despite complex ice motions, as is the case in our current dataset.

Better and more representative sampling of waves in ice is critical to help draw robust conclusions about the underlying physics and alleviate the risk of overfitting models to idiosyncrasies and noise in individual datasets. As an illustration of

this fact, the discussions in Kohout et al. (2020) disprove surprising findings previously reported in Kohout et al. (2014), and attribute these now disproved findings to considering only a few buoys for a short period of time under specific conditions. In a similar way, Thomson et al. (2021a) demonstrate that noise properties of close source buoys (which make noise characteristics harder to understand, quantify, audit, and detect) have caused spurious rollover in dataset, and cast doubts on the rollover observations of numerous previous studies, although this does not absolutely rule out that rollover could happen.

Therefore, the use of larger swarms of open source buoys opens up much needed better sampling of the MIZ. This, combined with open data release policies, enables new MIZ in situ studies, focused on the use of high-density, high-accuracy waves in ice measurements (Rabault et al., 2023; Nose et al., 2023b, a). This trend is now making it possible to improve the validation of models (Nose et al., 2023b), and will possibly also allow much needed further development of the geophysical transfer function to interpret satellite images (Liu et al., 2021; Collard et al., 2022; Shen et al., 2017; Horvat et al., 2020), as well as to improve the understanding of the physical processes happening in the ice. As a consequence, a number of campaigns are now deploying e.g. OMBs or microSWIFTS or other similar buoys en masse, allowing to generate significant amounts of data about regions which have traditionally been undersampled, and providing new inputs to sea ice scientists and modelers.

In the present paper, we focus on in situ data retrieved from an OMB deployment. The deployment took place in 2021, and contains complex patterns of waves in ice significant wave height (SWH) modulation. This is the first time, as far as we know, that such a drastic quasiperiodic modulation of the SWH is reported and specifically analyzed in the sea ice. We show that the evolution in the SWH very likely cannot be explained by neither classical mechanisms such as wave-current interaction, nor existing wave models including state-of-the-art waves in ice damping parameterization. Therefore, we conclude that this observation is due to the effect on the ice cover and is the signature of complex dynamics happening in the ice.

The structure of the manuscript is as follows. We first present the different sources of data and numerical models available to perform our case study. We then proceed to analyze the results of the models and in-situ data, investigating if these are able to reproduce the in-situ observations. Finally, we discuss possible explanations for the observed dynamics, and we both formulate a conjecture around the nature of candidate mechanisms that could produce the observed features, as well as suggest how this conjecture could be tested in further measurement campaigns.

## 2 Data sources

In this section, we present the different sources of data used in the analysis and discussion and show their main features.

### 2.1 Observation data sources

#### 2.1.1 Buoy data: direct observations of drift and 1-dimensional wave spectra

The core of the observations presented here is a series of waves in ice buoy trajectories and wave observations collected between February and March 2021 in the Barents Sea, South East of Svalbard, using OMB v2018 (Rabault et al., 2020) and v2021 (Rabault et al., 2022). In this deployment, the OMBs v2021 were still prototypes, and not all OMBs v2021 were equipped

with wave measurement hardware. These data are already openly released and presented in detail in Rabault et al. (2023), and we refer the reader curious of technical details to the corresponding publication and the cruise report (Nilsen et al., 2021). In summary, OMBs report sea ice drift tracks (as observed from GPS measurements) and waves in ice 1-dimensional spectra (as observed from Inertial Motion Unit (IMU) measurements), both transmitted in near-real-time over the iridium network.

The deployment of the OMBs used in this study took place in February 2021 in the Barents Sea East of Svalbard, with the deployment details reported in Nilsen et al. (2021). A synoptic view of the whole deployment is provided in Fig. 1. Although 17 buoys were deployed in total in a wider area, we focus on analyzing the data from a group of 4 buoys (with IDs 200913, 13319, 200905, 19648) that are clustered around the south-west tip of Svalbard. The other buoys were, at the time of the event we consider, either spread further away from the key area of interest, or OMB-v2021 prototypes without wave measurement ability, and their data are not considered here. We want to underline that while we focus on only a few buoys that are closely distributed in space in the following analysis, the reason why such a set of closely distributed buoys is obtained is that a larger set of 17 buoys were deployed initially, which drastically improves the probability to get a buoy cluster even several days to weeks after the deployment has taken place. The specific wave event that we focus on took place between March 1st 2021 and March 4th 2021. The SWH and trajectories reported by the 4 instruments we consider are presented in Fig. 2, and the wave spectra for the 3 buoys furthest into the MIZ (IDs 19648, 200905, 13319) are visible in Fig. 3. The buoy furthest out of the MIZ (200913) is only discussed quickly to provide a reference observation where no modulation is present, and in the following, we will refer to the buoys with IDs 19648, 200905, and 13319, as the "buoys of interest" (BOIs), which will be the focus of our analysis.

The dominating feature visible in Figs. 2, 3 is the strong modulation, with a period of around 12 hours, that is visible in the SWH and the 1D wave spectrum of the instruments inside the MIZ. This modulation is visible consistently across the 3 BOIs, that include 2 v2018 and 1 v2021 buoy models. Instruments situated on the very edge, or outside, of the MIZ do not display such modulated SWH patterns, as visible from the SWH timeseries from the instrument with ID 200913 (a similar conclusion is obtained from other buoys in the area, not shown here). This modulated SWH pattern is an outstanding feature in the present dataset, and the following analysis will focus on understanding this specific event. Although rapid increases in the SWH observed in sea ice have been observed in the past and are related, e.g., to the breakup of sea ice induced by propagating wave trains (Collins III et al., 2015), the present data are unique in that a periodic increase and decrease in the SWH is observed.

For BOI 19648, which is the deepest in the MIZ and has the highest temporal resolution (since it is an instrument v2021), a significant wave height $SWH_{T00}=0.33m$ is observed at 2021-03-02T00:00Z, before reducing to $SWH_{T06}=0.03m$ at T06:01Z, and increasing again to $SWH_{T12}=0.34m$ at T12:00Z. This represents a modulation ratio between the maximum and minimum SWH reported in 12 hours, $(SWH_{T00} - SWH_{T06})/SWH_{T00}$, of more than 90%. We note that the actual modulation ratio may be even higher than our analysis reveals, since (i) the 0.03m SWH reported at T06:01Z comes close to the noise background of the instrument, so that the SWH value reported then may include some level of noise and may be slightly higher than the actual SWH at the location of the instrument. Moreover, (ii) the time resolution of the SWH measurements for instrument 19648, though higher than for the other instruments, is still only every 2 hours, so it is likely that the "coarsely sampled" minimum (resp. maximum) measured does not match exactly the time when the actual minimum (resp. maximum) SWH is reached in

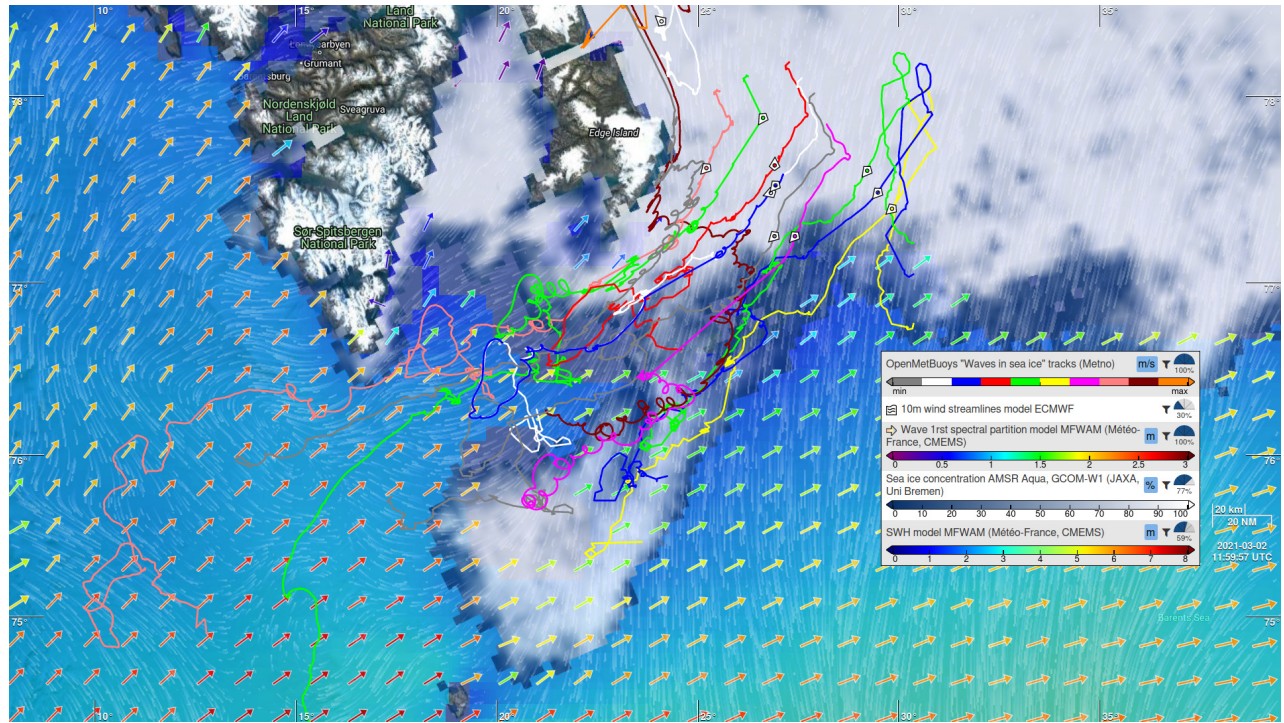

**Figure 1.** Synoptic overview of the conditions in the area of the buoys during the modulated SWH event. As visible, the buoys (and in particular the 3 BOIs - the reader is invited to browse the data online to individually select buoy trajectories) are well within the MIZ (the positions of the buoys taken closest to the reference time 2021-03-02T12Z are indicated with markers). Wind comes from the North and, due to the presence of the sea ice, contributes little to the local SWH. Swell incoming from the South-West is propagating through an open water opening before penetrating the MIZ and propagating to the location of the buoys. This figure is representative of the synoptic conditions over the time extent of the wave modulation event. The data can be viewed interactively online by following the permalink: https://odl.bzh/gVOsfRnq (accessed November 2023).

reality. The typical peak wave period in the outer part of the MIZ is around 12s, and at the BOIs it is around 14s through the
145 event, as visible in Fig. 3. These values are typical for records of waves in ice in the area.

A clear general drift pattern is visible on the GPS tracks of the buoys. All buoys drift generally to the south-west during the duration of the event, with additional motion taking place at a period of around 12 hours, likely due to a combination of inertial oscillation and tidal currents (both have around the same period in this area), as visible specifically for the BOIs in Fig. 4. This general sea ice drift motion is accompanied by strong shear and convergence / divergence motion in the sea ice. To
150 illustrate this, we analyze the triangle element obtained by combining the 3 BOIs and compute the associated divergence by (i) interpolating the position of the buoys on a common time base, (ii) using a first-order finite difference in time to compute an estimate of the velocity of the 3 BOIs on this common time base, and (iii) computing the local divergence following the methodology of Kwok et al. (2008):

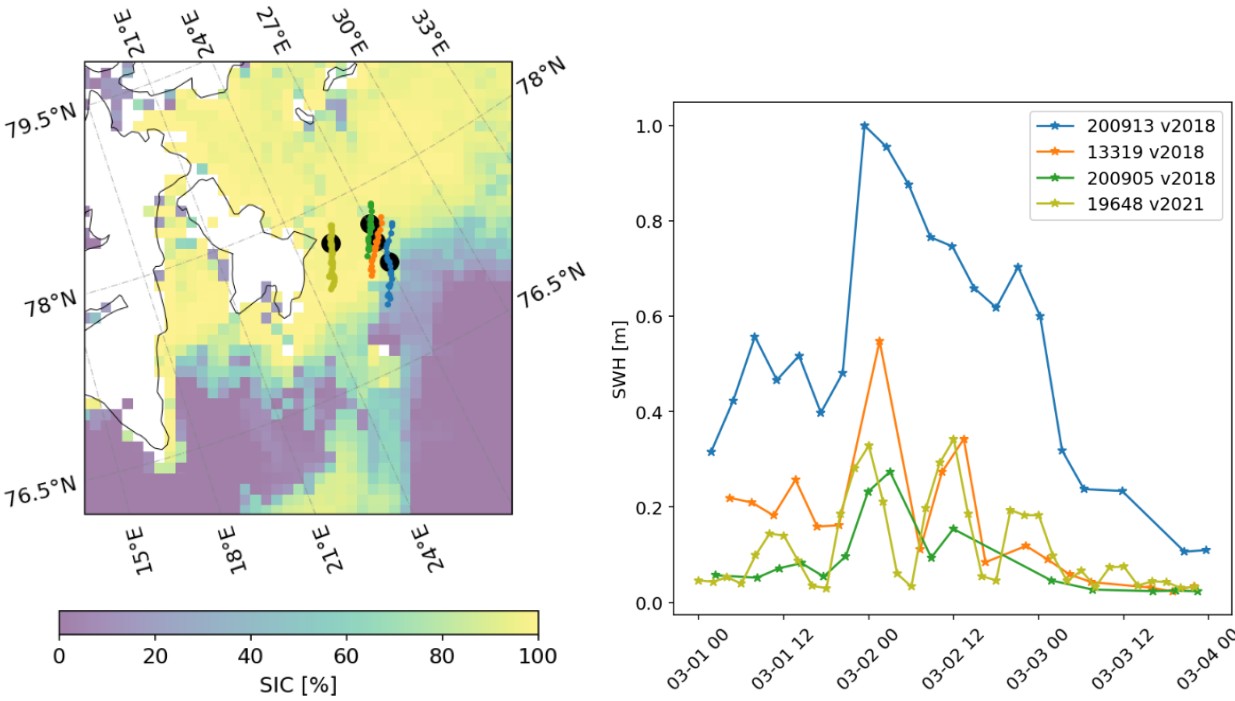

**Figure 2.** Overview of the buoys considered during the modulated significant wave height event. The overview map subfigure (left) presents the sea ice concentration (SIC, data source: internal product at the Norwegian Meteorological Institute, obtained from combining AMSR2 passive microwave observations and Sentinel-1 SAR observations) at 2021-03-02T06:00Z, around the middle of the time window of the SWH modulation event. The black dots indicate the position of the buoys at the corresponding time, and the trajectory for each buoy is indicated in a separate color, corresponding to the caption on the SWH subfigure. The SWH subfigure (right) shows the timeseries of the SWH. The instruments furthest into the MIZ show clear SWH modulation with a period of around 12 hours (instruments with IDs 19648, 200905, 13319 in particular, which we call "buoys of interest" (BOIs) in the following; these are the ones we focus on in the rest of this work). Instrument 19648, which is the deepest into the MIZ, displays a modulation ratio between the maximum and minimum SWH it observes over 12 hours of up to 90%. Instrument 200913, which is furthest outside of the MIZ, does not show a similar modulated SWH pattern.

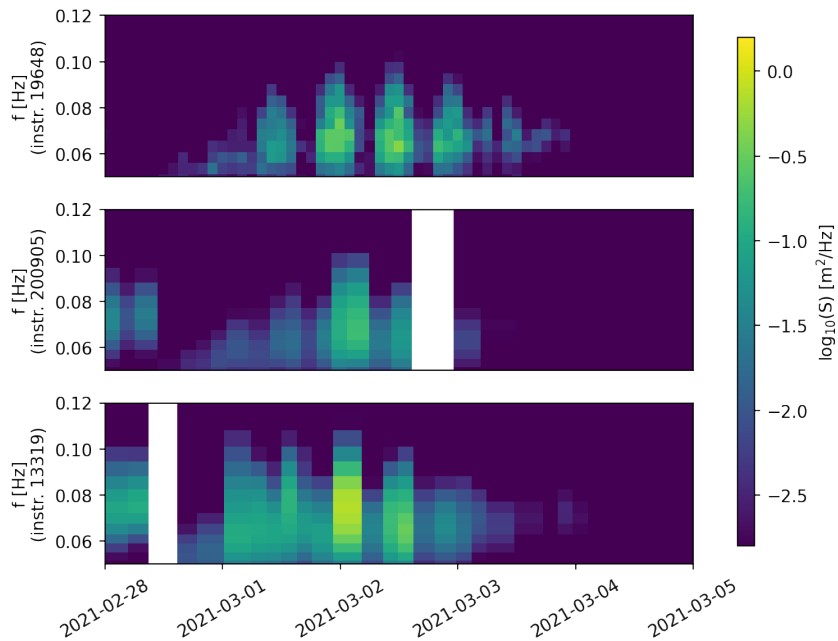

**Figure 3.** 1-dimensional wave spectrum for the 3 instruments furthest in the MIZ that are equipped with wave measurement capability (referred to as the Buoys Of Interest (BOIs) with IDs 19648, 200905, 13319), obtained during the time of the modulated SWH wave event. The instrument ID 19648 is a v2021 prototype, with higher wave measurement rate, a higher frequency resolution for the transmitted 1D spectra, and more sophisticated wave spectrum buffering and iridium transmission retry strategy than the 2 other v2018 instruments, resulting in better resolution of the wave spectra in both time and frequency and fewer missing measurements. The modulation in the wave activity is clearly visible on all instruments, and is particularly striking on the instrument ID 19648 thanks to its higher temporal resolution. The wave spectra shown here confirm that the SWH oscillatory event observed in Fig. 2 corresponds to an increase and decrease of the wave activity over the full frequency range, is observed consistently across the 3 BOIs, and is not the result of some measurement artifacts.

$$\nabla \cdot \mathbf{u} = u_x + v_y, \tag{1}$$

where $\nabla \cdot \mathbf{u}$ is the divergence of the velocity of the sea ice, with $\mathbf{u} = (u, v)$ the velocity of the sea ice in the east-west and north-south directions, and $u_x$ and $v_y$ are the spatial gradients in the motion of the BOIs calculated using a line integral along the boundary of the triangle:

$$u_x = \frac{1}{A} \oint u \, dy, v_y = -\frac{1}{A} \oint v \, dx, \tag{2}$$

where $A$ is the triangle area. Line integrals are approximated based on the motion of the BOIs at the edges of the triangle
element interpolated in time on the common time base, similar to Kwok et al. (2008). For example, we compute $\oint u\,dy$ as:

$$\oint u\,dy = \sum_{i=1}^{3} \frac{1}{2}(u_{i+1} + u_i)(y_{i+1} - y_i), \tag{3}$$

where the subscripts are cyclical (i.e., $u_4 = u_1$), and we have used centered finite differences for $u \approx (u_{i+1} + u_i)/2$ and
$dy \approx (y_{i+1} - y_i)$, similar to Kwok et al. (2008). Following Kwok et al. (2008), this results in a trapezoidal rule for the linear
interpolation of $u$ between BOIs. Similar formulas can be written for the other partial derivatives, and the area $A$ is computed
as:

$$A = \frac{1}{2} \sum_{i=1}^{3}(x_i y_{i+1} - y_i x_{i+1}). \tag{4}$$

The algorithm and code used for the computation of the divergence are available in a notebook on the GitHub Data and Code
Release page. The divergence coefficient obtained is presented in Fig. 4. Strong patterns of sea ice convergence and divergence
are observed, consistent with previous reports in the same area (Marchenko et al., 2011). As visible in Fig. 4, there is a clear
correlation between the modulation observed in the SWH and the local divergence rate. This will be discussed in detail later in
the manuscript.

### 2.1.2 Sea ice imaging using SAR satellite data

We also look directly into satellite images available over the area to further visually confirm, based purely on direct satellite
observations, that the sea ice conditions correspond to the models and that the instruments are well inside the MIZ during the
event. An overview of the buoys and sea ice conditions observed directly from Synthetic Aperture Radar (SAR, gamma-0 from
Sentinel-1 in EW mode (Torres et al., 2012)) is provided in Fig. 5. As visible there and in good agreement with the figures
presented above, the BOIs are well within the MIZ over the duration of the modulated SWH event (the reader is invited to
browse the online data, where individual trajectories can be selected).

The SAR images can also be used to cross-check the kind of sea ice conditions locally present in both the outer MIZ and
around the BOIs. Several SAR image zoom-ins are presented in Fig. 6, both in the outer MIZ (subfigures (a) and (b)), at the
location of BOI 19648 (subfigure (c)), and much further north (subfigure (d), used as a reference to illustrate conditions with
large unbroken ice floes). The broader perspective SAR images visible in Fig. 5 can be used as a reference to understand
the meaning of the colors in Fig. 6 (observing the wider area is necessary to understand the meaning of the grayscale colors
in the figures, as the grayscale color depends on the SAR mode and polarization, as well as the postprocessing applied). In
particular, it is clear from Fig. 5 that given the polarization and post-processing used with the specific swath of data considered,
the "whiter" areas correspond to ice and the "darker" areas correspond to water. Fig. 6 confirms that waves propagate from
the South-West direction over the ice tongue south-west of the BOIs (wave crests in the ice are visible with the naked eye)

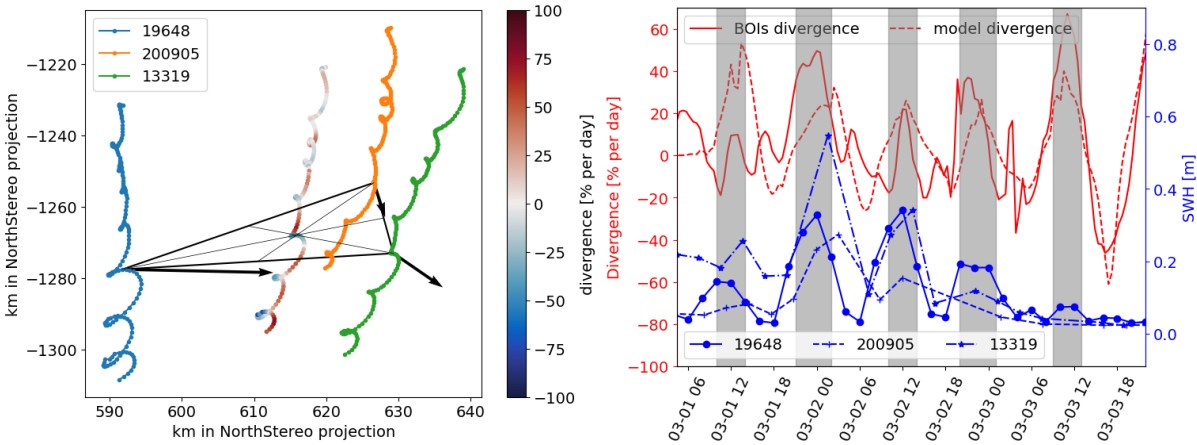

**Figure 4.** Details of the drift pattern observed for the 3 BOIs. Left: zoom-in onto the drift trajectories for the BOIs over the period 2021-02-29 to 2021-03-03. A clear drift pattern from North to South is observed. In addition, higher frequency motion (with a period of around 12 hours) is visible. We also show the barycenter of the 3 BOIs, which we color by the instantaneous value of the motion divergence, computed based on the triangle element formed by the 3 BOIs. The black triangle and arrows illustrate one instantaneous triangle element, its associated barycenter, and the velocity of the 3 BOIs at the corresponding time. Right: illustration of the instantaneous divergence rate computed for the BOIs triangle ("BOIs divergence" curve), alongside the SWH measured by the BOIs; a clear correlation is observed. The locally averaged sea ice divergence from the sea ice numerical model (curve "model divergence", see section 2.2.3) is also included, and shows good agreement with our in-situ measurements, which is a cross validation of the sea ice model quality.

and that ice conditions are most likely dominated by broken ice floes pressed together and refrozen areas (no large-scale consistent floes can be observed). This is especially visible in comparison to the image obtained much further north (Fig. 6 (d)), where, by contrast, large individual ice floes are observed. Therefore, this indicates that the sea ice around the BOIs is most likely constituted of broken ice floes with dimensions typically smaller than or at most comparable to the wavelength, pressed together, and that the sea ice likely does not behave like a single continuous solid plate, similar to what is reported, e.g., by Sutherland and Rabault (2016) when cracks and discontinuities are present in the ice cover. This is also consistent with the observation of strong sea ice convergence and divergence patterns. Indeed, in order for the ice cover to be able to sustain large divergence and convergence, there must be enough areas of open water leads that can open and close in order to sustain the "compression", respectively, "extension", of the ice cover.

## 2.2 Model-based data sources

### 2.2.1 Synoptic view of the ocean and atmosphere conditions in the area of interest

A synoptic view of the situation at 2021-03-02T12Z was presented as a part of Fig. 1. To make the synoptic view as interactive as possible, and offer more exploration freedom than what is allowed by presenting a few static plots, the view is generated using the Ocean Virtual Laboratory (OVL) online tool, and the users can follow the permalink https://odl.bzh/gVOsfRnq

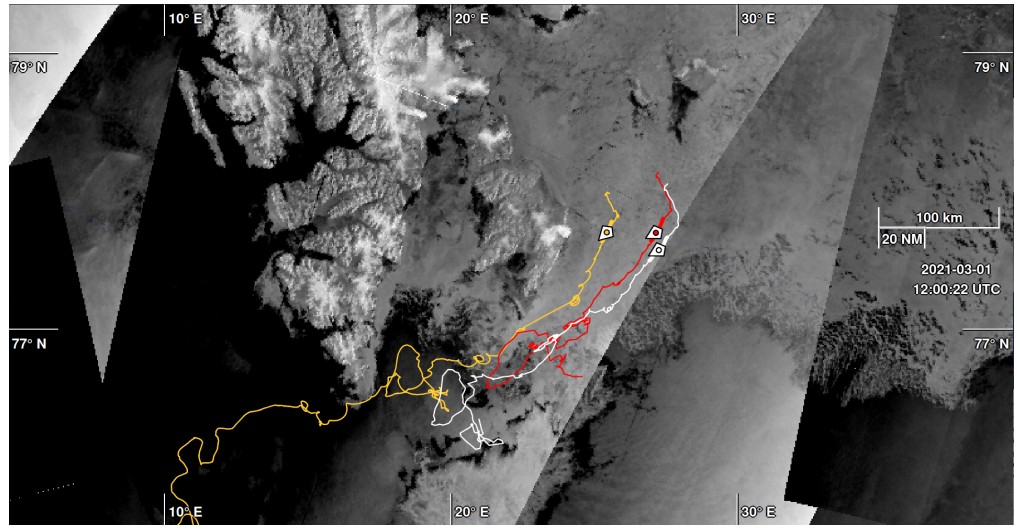

**Figure 5.** SAR overview of the 3 BOIs locations East of Svalbard at the reference time 2021-03-01T12Z. There are more buoys in the area than just the 3 BOIs considered in the present study, and the reader is invited to browse the online data, where individual trajectories can be selected. Sea ice is clearly visible from Sentinel-1 SAR gamma-0 EW observations (the time width for collocation matching is taken equal to 1 day). Each line indicates an OMB trajectory, and the position of each buoy at the reference time is indicated by a marker. As visible on the SAR image (more information can be obtained by following the link below and interacting online with the data), the BOIs are well within the MIZ, in the middle of an area of broken ice floes. The illustration is generated using the Ocean Virtual Laboratory online viewer (Collard et al., 2015), and it can be re-generated and explored interactively using the permalink: https://odl.bzh/jG7KkYvJ (accessed June 2025).

(accessed June 2025) to access the same map and be able to move forward and backward in time, as well as switch on and off different map layers. The map presented in Fig. 1 shows the Svalbard area where the buoys are drifting, together with layers reporting (i) interactive markers representing the wind, (ii) the local sea ice concentration obtained from the ARTIST Sea Ice (ASI) algorithm on the AMSR2 sensor of the JAXA GCOM-W1 satellite (data provided by Institute of Environmental Physics, University of Bremen (Spreen et al., 2008; Melsheimer and Spreen, 2019)), (iii) the total significant wave height obtained from the global wave model MFWAM of Météo-France with ECMWF forcing and satellite data assimilation (Aouf et al., 2019; Dalphinet et al., 2022), (iv) the direction and SWH associated to the swell first partition, also obtained from MFWAM. We encourage the reader to view the synoptic map interactively online following the permalink provided above to observe the conditions at neighboring times.

As visible in Fig. 1, the permalink, and in good agreement with Fig. 2 and Fig. 5, the BOIs are well into the ice area during the SWH modulation event. Local wave conditions are dominated by incoming swells, originating from the South West, which propagate through an open-water opening before penetrating the sea ice. The wind comes mostly from the north, which is covered by sea ice, and therefore we expect little locally generated wind wave contributions in the sea ice. The map is generally representative of the situation over the time extent of the SWH modulation event reported, though the SWH of the

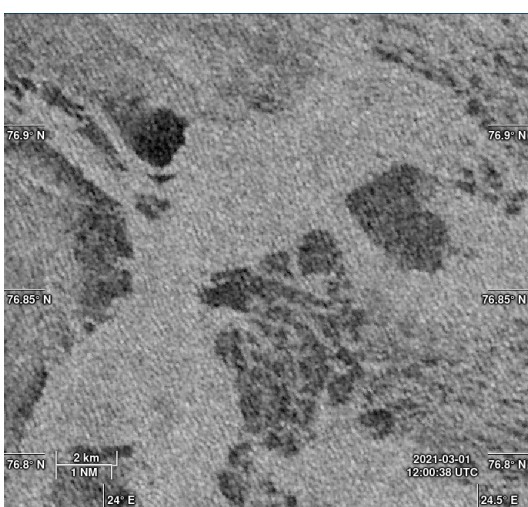

(a) SAR image from the outer MIZ, over the ice tongue South West of the BOIs between Hopen and Svalbard, through which the dominant incoming swell waves travel on their way to the BOIs. Permalink: https://odl.bzh/oewZ8UtD .

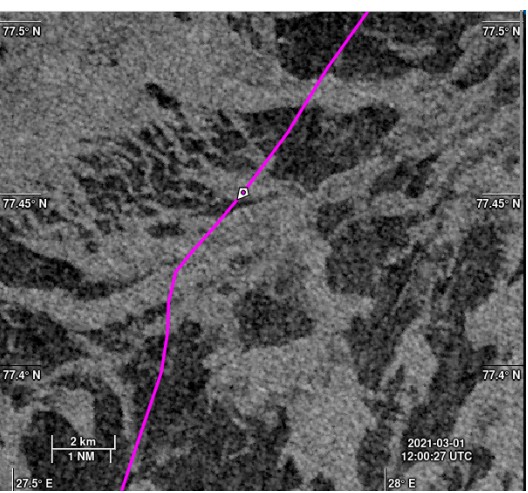

(b) SAR image from the outer MIZ, in the neighborhood of buoy 19643 (marker indicates position at 12UTC), South from the BOIs. Permalink: https://odl.bzh/0j3fBx8k .

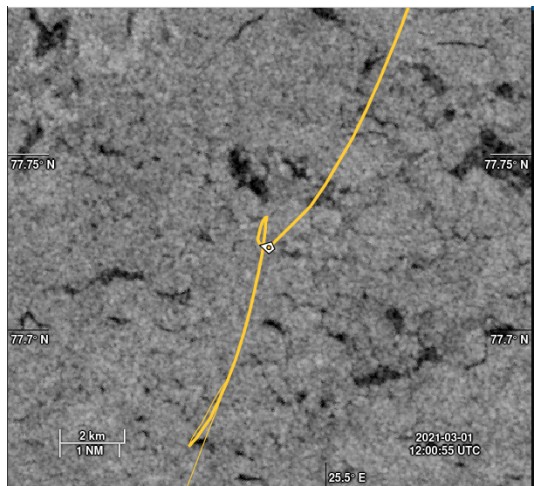

(c) SAR image from the area around BOI 19648. The position of the BOI 19648 at 12UTC is indicated by the marker. Permalink: https://odl.bzh/JBYRiR6- .

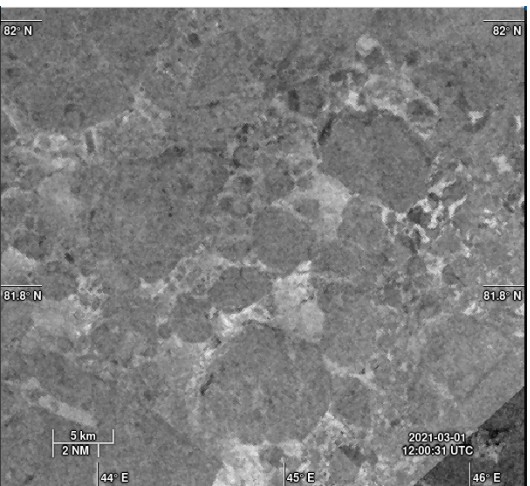

(d) SAR image far North from the area of the BOIs, where clear unbroken ice floes with typical size O(1-10km) are present. Permalink: https://odl.bzh/M1wrLH_v .

**Figure 6.** Selection of illustrative SAR images zoom-ins acquired on 2021-03-01 at around 12UTC in the general area of the buoys (except for subfigure (d), which is much further North). The outer MIZ (subfigures (a) and (b)) presents a mixture of areas of broken floes and open water leads. The relatively high sea ice concentration and the limited extent of the open water leads and refrost areas implies that the fetch available for wave development under the influence of wind quickly becomes negligible into the sea ice. Waves are clearly seen propagating from the South West over the ice tongue (subfigure (a)), as also observed in the models (see sections below), confirming the origin of the waves observed. By contrast, the ice concentration around instrument 19648 (subfigure (c)) is higher, however, many smaller leads, cracks, refrozen areas, and a lot of complex structure is visible in the SAR images. This indicates that the sea ice around instrument 19648 in particular, and the BOIs in general, is not constituted of large homogeneous floes, but of many medium size broken floes. The fact that there are no large unbroken floes in the neighborhood of the BOIs is especially visible when comparing to areas much further North (subfigure (d)), where large unbroken ice floes separated by a matrix of smaller broken ice bits and refrozen areas are clearly visible.

incoming waves varies in time (but with a much slower typical SWH variation timescale and without the 12-hour modulation we observe in the sea ice), and the direction of propagation of the swell first partition shows minor fluctuations.

Sudden, large-amplitude changes in the temperature of the ice, usually caused by changes in the air temperature above, can also influence the mechanical properties of sea ice and the wave in ice propagation and damping. Therefore, we also consider the 2-meter temperature in the area, as provided by the ERA5 reanalysis (Hersbach et al., 2020). We could observe (not reproduced for brevity) only moderate temperature changes in the atmosphere over the duration of the modulated SWH event, typically within 4 degrees, and no periodicity is visible in the temperature signal.

### 2.2.2 Custom wave model data in the MIZ during the oscillatory event

In order to investigate whether wave-current interaction, local wind-wave generation, and established wave-in-ice attenuation mechanisms, are able to explain the oscillatory damping we observe, we run a series of custom local spectral wave models in the area of interest. The wave model used is a third generation WAM Cycle 4.7 developed at Helmholtz-Zentrum Geesthacht (Group, 1988; Günther et al., 1992) with modifications made at the Norwegian Meteorological Institute, Norway (MET-Norway). These modifications consist of allowing the propagation of waves under sea ice following wave attenuation (Yu et al., 2022), and a correction of wave growth in very high winds (Breivik et al., 2022).

The parameterization from Yu et al. (2022) is chosen as we have observed experimentally in several studies in the past years, both at the Norwegian Meteorological Institute and at MeteoFrance, that this is the one that performs best without ad hoc tuning. This was briefly discussed in a report (Bohlinger et al., 2024) and a proceeding (Aouf et al., 2024), and has been independently confirmed by model runs of both WAM and WW3 performed by coauthors of this work from both institutions. Although this may seem surprising at first, since the model from Yu et al. (2022) was developed from data from the Southern Ocean Autumn, this may indicate that the same general viscous dynamics play a role in a wide variety of sea ice conditions.

Wave propagation in ice-covered areas is modeled by weighting the energy source and sink terms used in the spectral model by the relative sea ice concentration. For example, the sink term used in the model to represent wave in ice damping is weighted by the local sea ice concentration value (that is, $\text{SIC}_{\text{local}}(x, y)$, the sea ice concentration expressed as a fraction between 0 and 1), while source terms arising from winds are weighted by $1 - \text{SIC}_{\text{local}}(x, y)$. Similarly to the findings obtained in the case of sea ice drift by Sutherland et al. (2022), this allows for a smoother and more realistic representation of the MIZ, compared to what is obtained by setting an arbitrary threshold separating areas where a purely open water vs. sea ice model parameterization is used. The spatial resolution of the model is 2.5 km. The spectral resolution is 24 directions and 30 frequencies with the first frequency equal to 0.034523 Hz. The boundary conditions for the wave spectra come from two sources: the European Center for Medium Weather Forecast when sea ice is included as forcing, and a coarser domain with no ice run at MET-Norway otherwise (Bengtsson et al., 2017; Batrak et al., 2018; Röhrs et al., 2023). The forcing fields used are surface winds from Arome Arctic (Müller et al., 2017), sea ice concentration and thickness from CICE5 (Bailey et al., 2018), and ocean surface currents, including tides, from ROMS (Shchepetkin and McWilliams, 2005).

The domain of interest around the buoys is a coastal area with strong currents and tides, and a natural candidate to the observed modulation could be the effect of wave-bathymetry, wave-current, and wave-tide interaction. Moreover, there are

250 open water regions south of Svalbard, so locally generated wind waves could also be a potential candidate explanation. Finally, the sea ice in the area is moving under the influence of tides, large-scale currents, and winds, which could participate in changing the wave in ice propagation distance and the resulting wave height observed at the location of the BOIs. In order to take into account these different effects and to investigate the impact of each physical mechanism on the local sea ice dynamics, either in isolation or in combination with each other, different model flavors are run, which we will refer to in the following as:

– W: including the locally generated wind waves, but not the effect of wave-current interaction nor the sea ice,

    – WC: including both local wind wave generation and wave current interaction, but not the effect of the sea ice,

    – WI: including both wind wave generation and the effect of the sea ice, but not the effect of wave-current interaction,

    – WCI: including all 3 effects, i.e. wind wave generation, wave-current interaction, and the sea ice.

The two-dimensional directional wave spectra from the model runs in these different configurations were stored along the
260 tracks of the BOIs.

When more information about sea ice is needed, we use data from the CICE5 model directly, which is run operationally at MET-Norway in the context of the Barents2.5 model and combines the CICE and ROMS models to simulate the dynamics of the sea ice sheet (Fritzner et al., 2018, 2019; Duarte et al., 2022). Bathymetry information is included in these models and taken into account in resolving wave propagation physics. In particular, Barents2.5 uses a 2.5km smoothed bathymetry map
(to keep the numerics stable).

### 2.2.3   Numerical model of sea ice dynamics

We have already highlighted in Fig. 4 that strong convergence and divergence are present in the sea ice and that these seem to correlate well with the observed SWH modulation observed at the location of the drifters. The existence of significant patterns of sea ice convergence and divergence dynamics is confirmed by visual inspection of maps of sea ice convergence
from the CICE5 model mentioned above, see Fig. 7. In particular, Fig. 7 confirms that the divergence and convergence of sea ice observed from the buoy trajectories are present in the whole area and show consistent patterns in space and time. Moreover, we observe that the dynamics and periodicity of the pattern oscillation between a strong state of convergence and divergence match the 12 hour periods observed in the SWH modulation, corresponding to a dipole-like structure propagating in the sea ice.

In order to further compare these model observations with the BOIs data, the sea ice divergence from the CICE5 model is extracted at the position of the barycenter of the 3 BOIs shown in Fig. 4 (left). The numerical model has some level of noise in both space and time, so we then computed the average between the divergence obtained for the neighboring grid points around the location of the barycenter (taking points that are in a box of +- 1 grid cell around) to get a smoother estimate. The obtained sea ice convergence is part of Fig. 4 (right). As visible there, the model data agree well with the value of the sea ice divergence
derived from the observations.

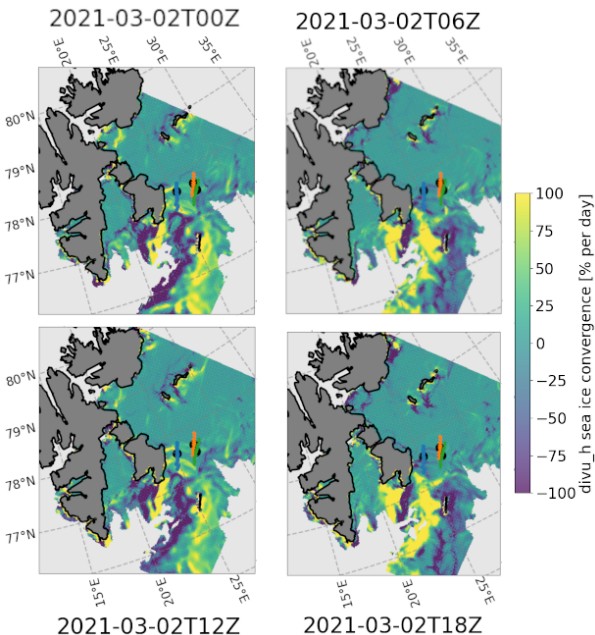

**Figure 7.** Illustration of the sea ice convergence and divergence pattern in the area of the buoys, as obtained from the sea ice velocity divergence field produced by the CICE5 model output. A clear pattern is visible both in space and in time, with the same 12 hours periodicity as observed in the SWH modulation. Further vizualization of the animated video of the sea ice model divergence shows the existence of a dipole that rotates in the neighborhood of the location of the buoys, leading to a periodic transition between phases of sea ice convergence and divergence at any point of the area considered.

Although the main point of this paper is not to study the sea ice convergence and divergence per se, experiments with and without tides have been performed in the MetROMS model data used to force the CICE5 runs (not reproduced here). Without tides, only weak sea ice convergence and divergence undulations of much smaller magnitude, with varying frequency and phase, are obtained. This confirms that the observed convergence and divergence of sea ice is most likely caused by tidal currents.

## 3    Model data analysis: what role is played by wave-current interaction, compared to wave-ice interaction, in the modulation observed by the buoys?

In this section, we investigate what categories of physical processes can explain the waves in ice SWH modulation that we observe. In particular, two categories of mechanisms are candidates for explaining our observations, as previously highlighted.

– M.1: Wave-current, wave-bathymetry, and wave-tide interaction: These mechanisms are well known for modulating the SWH in areas where strong tides and tidal currents are present. This category of mechanisms is independent of the presence or absence of sea ice in the area and should be observed both in open water and in ice-covered conditions.

- M.2: Wave-ice interaction and wave attenuation by sea ice. If this category of physical mechanisms is responsible for the observed SWH modulation, the effect of sea ice should be a key ingredient in reproducing the modulation observed, and such a modulation should not be observed without the sea ice. This does not mean that the currents and tides do not play a role: for example, the currents and tides may be the physical forcing that causes the changes in sea ice behavior, which in turn explains the modulation observed. However, in such a case, it would be the sea ice, not the tides and currents per se, that would be causing the physics at the origin of the wave attenuation and SWH modulation observations. As a consequence, similar modulations would in this case not be observed under ice-free conditions.

Our goal in this section will consist of estimating the likely importance and variability of mechanisms belonging to M.1 and M.2 for which we can produce a quantitative estimate that could be involved in causing the modulation we observe. We will perform such an analysis relying on both the in situ data collected and the numerical simulations and models that are available in the area.

We note that the physics at play in both M.1 and M.2 are highly complex, particularly in the Barents Bank, where complex bathymetry, tides, and currents are present. Therefore, the following analysis will necessarily rely on a number of approximations and simplifications, and none of them, taken individually, is a compelling conclusive proof. However, our aim is to compare different approaches and check if they arrive at the same typical conclusions and order-of-magnitude estimates.

We also want to highlight that some potential mechanisms are well established and can be quantified, while some others are more challenging to estimate. For example, regarding possible explanations belonging to the M.1 category, the models should provide reasonable estimates of wave-bathymetry and wave-current interaction, including the refraction of waves by current gradients, since these are the result of complex but well established physics. Similarly, in the M.2 category, the effect of MIZ edge displacement under the influence of strong currents and winds, and hence increased wave in ice attenuation due to increased wave propagation distance in the sea ice, should be well captured by ice models corrected with assimilated satellite measurements. However, some other mechanisms, like a hypothetical change in the wave in ice energy dissipation following modifications of the sea ice state of divergence and "closedness", cannot be readily estimated from current models.

Therefore, the following analysis can either reveal that a known mechanism belonging to the M.1 or M.2 category is able to explain for the SWH modulation observed at the BOIs, or, if unsuccessful at explaining the observations, that additional physics must be considered.

## 3.1 Numerical models analysis of the effect of bathymetry, currents, and tides on significant wave height

Our objective in the present section is to determine whether wave-current interaction or simple sea-ice dynamics (such as sea-ice drift leading to a change in the wave in ice propagation distance) can explain the modulation observed, according to advanced, state-of-the-art numerical model runs that include the wave-current interaction and estimates of the sea-ice attenuation. Compared with the wave-ray analysis discussed in Appendix A, these models include additional physics (such as nonlinear interactions and dynamics of the full wave spectrum during wave propagation).

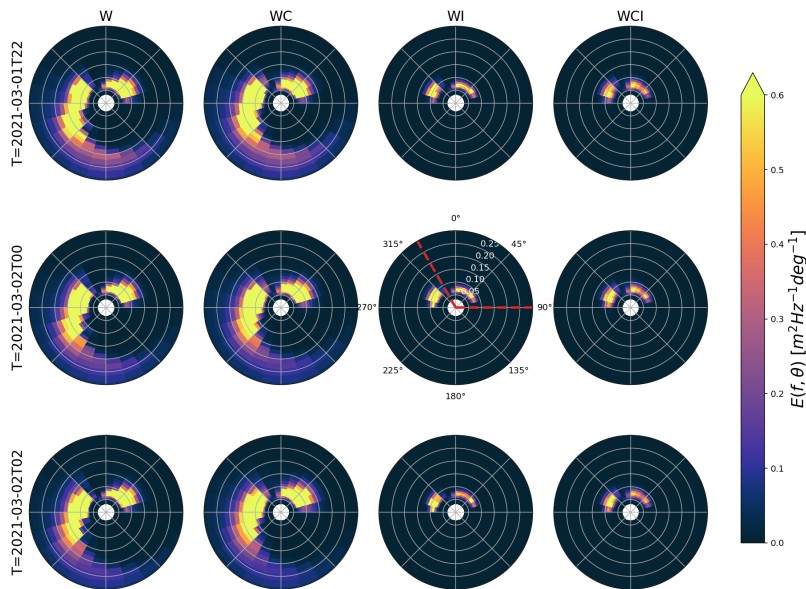

**Figure 8.** Directional spectra from the custom WAM model runs. Columns from left denote W, WC, WI, and WCI model runs, respectively, while rows from top denote the temporal evolution with steps of 2hrs following the track of the BOI 19648. Red lines in the middle panel in the WI column denote the sector for the swell components that undergoes most tidal modulations due to currents, and are thus used for the spectral partitioning "S1".

A subset of the directional spectra obtained from the different model runs along the trajectory of BOI 19648 are presented in Fig. 8. The sea state was bimodal for all the wave model runs. Compared to the results presented in Fig. 1, the dominant main swell propagating from the South-West towards the North-East is well reproduced. In addition, the model reveals a secondary lower amplitude slightly higher frequency swell signal propagating toward the north-west, which was not visible in Fig. 1, since only the dominant swell component was shown there. The shorter wind-wave frequencies are, as expected, not present in

the model runs including sea ice effects, due to both the attenuation effect by the sea ice, and the fetch blocking effect of the sea ice that prevents wave growth in ice covered areas since the wind is generally blowing from the sea ice towards the open ocean during the period considered. In order to quantify the current-induced modulation on the different components in model runs without sea ice, we perform a spectral partitioning in direction, since the dominating directions were more or less stationary throughout the study period (not shown here). The swell components within the range $(330°, 90°)$ (sector indicated by the

red limits on one of the directional spectra at Fig. 8) are hereinafter denoted as "Swell 1", and will be used when analyzing model outputs without sea ice included (i.e., model outputs from W and WC runs), since the dynamics of the swell-current interactions of interest here are otherwise hidden in the locally generated wind wave signals that are forced in the models in the absence of ice.

The temporal evolution in SWH obtained by integrating over the directional spectra, along the position of the BOI 19648 (which is furthest into the sea ice and shows the strongest modulation) where these are taken, for the W, WC, WI and WCI cases, are presented in Fig. 9. For cases without sea ice, only the Swell 1 partition is considered as discussed above. For cases with ice, the whole directional spectrum is considered. Several interesting aspects are visible there.

First, the Swell 1 partition in the models without sea ice (W and WC) predicts a much too high SWH at the location of the instrument, as expected. If we add the wind sea and secondary swell component, the SWHs for the models W and WC would be about 1–2 m higher than what is observed at the BOI 19468 throughout the period (not shown). The higher SWH obtained in model runs without the sea ice is of course expected, as the sea ice induces a strong energy dissipation, resulting in attenuation of the waves. This effect is naturally not included in the W and WC cases.

Second, model runs with included sea ice attenuation (WI and WCI runs) produce qualitatively reasonable SWH values in the MIZ at the location of the instrument, except for the "modulated" attenuation effect that is not reproduced. This is interesting on two aspects: first, it indicates that the wave in ice attenuation parameterization used, which was not tuned specifically for the present case but used default values from Yu et al. (2022), is reasonably effective at capturing the typical intensity of the wave in ice attenuation effect, and the resulting SWH at the location of the BOI 19648. Second, it shows that, while the case we study displays strongly modulated SWH values, the modulation we observe is likely an additional effect that comes on top of the wave attenuation mechanisms that are robustly captured by the established attenuation parameterizations.

In order to estimate the impact of the wave-current interaction on the SWH predicted by the models, we computed the relative difference between model runs with and without wave-current interaction, defined as:

$$R_{WC(I),W(I)} = \frac{SWH[WC(I)] - SWH[W(I)]}{SWH[W(I)]}, \tag{5}$$

where the case pairs without (WC vs. W) and with (WCI vs. WI) the ice effect are considered. We find that the modulations calculated for SWH are mostly between 10 %–30 % for both W against WC (using the Swell 1 partition), and WI against WCI (upper panel Fig. 9). Such relative differences are similar to what is reported elsewhere in terms of the intensity of current-induced modulations (Ardhuin et al., 2017; Halsne et al., 2022). This 12h modulation is less evident when considering the wind-sea part as well as the full spectrum in the WC vs. W case, indicating that the swells propagating across long distances and that do not receive a locally generated energy input are the most affected by the current modulation effect.

Finally, since (i) we observe that no or only little modulation is present in the WI and WCI cases, and (ii) the sea ice input is provided by the CICE5 model, which includes sea ice drift, melting, and freezing, as simulated by a numerical model and corrected by assimilated satellite data, we conclude that simple sea ice changes, such as large displacements in the MIZ edge, or rapid changes in the sea ice thickness, are unlikely to be the cause of the SWH modulation we observe. Indeed, these effects should be, if not perfectly, at least reproduced overall by the WI and WCI model runs.

These results suggest that, according to the numerical model runs, the observed SWH modulation observed by the buoys, which shows a modulation ratio of up to 90%, as previously highlighted, cannot be explained by the wave-current interaction (which is predicted to be at least 3 times smaller even when it is at its strongest), nor by simple sea ice dynamics included in

present sea ice models and wave in ice attenuation parameterizations. Moreover, the modulation observed in the sea ice due to the wave-current interaction is only around 5-10% when the strongest modulation ratio is observed from the buoys around 2021-03-02T00:00 to T06:00, which is an order of magnitude weaker than the modulation measured by the buoys. Although this is not an absolute proof that wave-current interaction cannot explain the observations we report, since numerical models are not perfect, this is an evidence going in this direction. The wave-current modulation obtained in the models, which is typically 10–30% at most, is also generally in agreement with what has been observed at other locations away from the shore, where strong tidal currents are present (Ardhuin et al., 2017; Halsne et al., 2022).

## 3.2 Investigation of current-induced wave field modulations using the wave action and wave ray approximations

Although full spectral wave models provide the most detailed analysis of the wave propagation and attenuation in the MIZ, they also introduce a lot of complexity. Simpler and robust approaches can also be used to cross-check the results. In particular, current- and bathymetry-induced wave refraction can be analyzed using the wave-ray method. We have performed such an analysis, which is described in detail in Appendix A. The results of the wave-ray analysis agree well with the results obtained from the spectral wave model, and confirm that the SWH modulation observed is most likely not the direct consequence of bathymetry and currents on the wave propagation.

## 3.3 Analysis of significant wave height satellite observations in ice free conditions, and correlation with tidal signal

An additional possible method to check that wave-current interactions are unlikely to cause the modulations observed is to consider the correlation between SWH in the area as measured by satellites when no ice is present, and tidal currents. For brevity, the technical details of this analysis are reported in Appendix B. There we conclude that, while it is well known that very strong SWH modulation can be obtained locally for very specific cases due to currents and bathymetry (see, e.g. Halsne et al. (2022); Saetra et al. (2021); Halsne et al. (2024)), this is most likely not the explaining factor for the modulation reported in the present case.

## 4 Discussion: what wave-ice interaction physics and mechanisms are likely explanations for the waves in ice modulation observed?

Our detailed analysis including a combination of numerical model runs including wave-current interaction, wave-ray paths, and a correlation analysis between local tides and SWH satellite observations when the ice is absent from the area indicate that the tidal current can probably only explain for a SWH modulation ratio typically at most around 30%, well below the modulation ratio of around 90% that we obtain in our in situ buoy observation timeseries. While numerical models are not perfect, we would expect state-of-the-art models to capture a much larger part of the intrinsic wave-current interaction if this was the dominating factor, especially as the 3 BOIs are spread over a domain about 50 km wide, so that the modulation we report can be seen on a wide area that covers several grid points at the model resolution of 2.5 km, rather than a very localized small-scale effect. Moreover, if the observed phenomenon was due to one of the intrinsic effects of tide and tidal currents

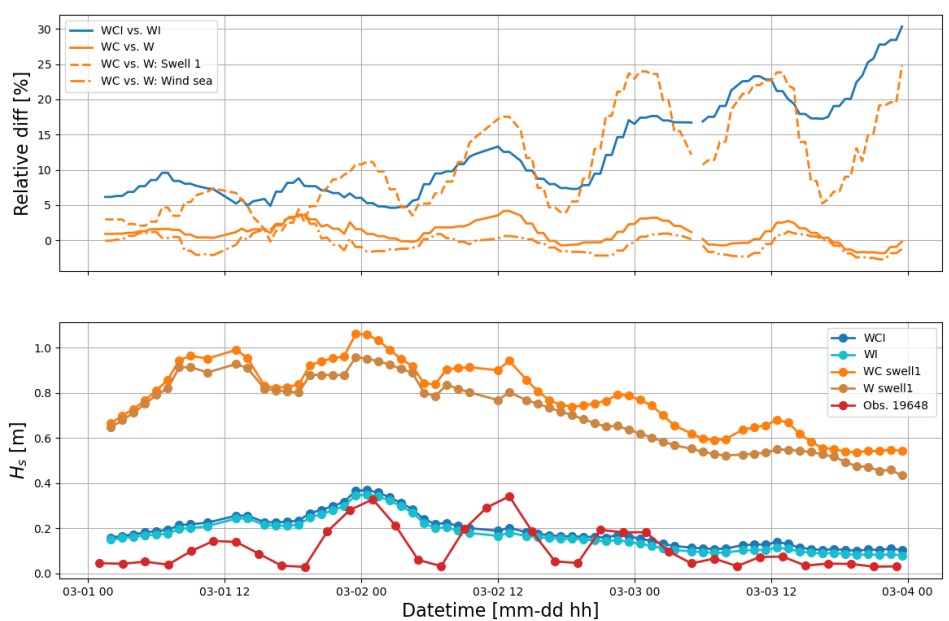

**Figure 9.** Temporal evolution in SWH (lower panel), and the relative difference due to the effect of currents (upper panel), for the WAM wave model runs. Different model runs, with (if the letter is present in the run case) or without Wind energy input (W), Current effects (C), and Ice effect (I), and combinations of these, are presented following the track of the instrument ID 19648, which is the instrument deepest into the MIZ, and has highest time resolution. Relative differences are computed using Eqn. (5) for WI vs. WCI (blue line), WC vs. W (orange line), Swell 1 of WC vs. W (dashed orange line), and the wind sea component of WC vs. W (dash-dotted orange line). The WAM model is able, overall, to reproduce the envelope of the SWH attenuation by the MIZ. However, the wave in ice modulation effect measured by the buoys, with an observed modulation ratio of up to 90%, is not reproduced by the model runs. The effect of currents, both with and without ice, is limited to a modulation of the SWH within 5–30% of the no-current value (W vs. WC, and WI vs. WCI runs). During the strongest modulation phase (2021-03-02T00:00 to T 06:00), a modulation ratio of 90% is observed by the buoy 19648, while the wave-current interaction computed by our model induces only a 5–10% modulation. Note that the strong increase in the relative difference at the end of the timeseries may, to some extent, be amplified due to the reducing baseline SWH value, which corresponds to a reduction of the denominator in Eqn. (5).

mentioned above, it could be expected to take place whether or not sea ice is present in the area. However, the correlation analysis between satellite SWH observations in ice-free conditions and tidal currents only explains for a modulation typically equal to what is predicted by the numerical model runs (which, incidentally, is a cross-validation of the accuracy of the tidal and wave-current modeling used).

Therefore, it appears that mechanisms independent of the presence of sea ice and its damping effect on sea ice consistently fail to reproduce the observations from the BOIs. As a consequence, we deduce from these evidences that the modulation we observe is likely related to the effect of the sea ice on wave propagation, and more specifically to wave in ice attenuation. However, while the model runs with sea ice attenuation successfully reproduce the typical envelope of the observed SWH, the modulation effect is not visible there either. The wave model simulations performed with sea ice take into account the sea ice conditions provided from a complete sea ice model that includes time dynamics of the sea ice cover, including changes in the sea ice location, concentration, and thickness. In particular, Fig. 4 confirms that good agreement is obtained between the sea ice divergence measured from the buoys and the one obtained from the CICE model. Other quantities, such as sea ice concentration and the associated "closedness" of the ice, cannot be directly measured by the buoys and compared to the model. However, these are closely related to the sea ice divergence, so these must most likely be represented well enough by the model to obtain the reasonable value obtained for the divergence. Therefore, we expect that, if these dynamics were the origin of the modulation we observe, this effect would be (at least imperfectly) visible in the wave-model output.

Since this is not the case, this is strong evidence that the modulation observed cannot come from simple effects such as changes in the sea ice cover or changes in the shape of large ice tongues leading to changes in the effective wave in ice propagation distance. This is further confirmed by a visual inspection of the SAR satellite images of the area over the duration of the modulation event, which confirms that the instruments are evolving relatively deep into the MIZ, and that no drastic changes or surprising features are observed on the ice cover as a whole over the duration of the SWH modulation event. Moreover, the timescale of the modulation event, which shows a period of 12 hours and a total duration of around 3 days, would anyway limit the amount of, for example, ice melting or freezing that can take place. In addition, the WCI model run successfully reproduces the bulk attenuation and the general envelope of the SWH observed at the BOIs, indicating that the attenuation mechanisms included in the WCI model successfully represent the dominating wave in ice attenuation contribution. Therefore, we conclude that the WCI model is successfully representing a range of physical mechanisms but that there must exist a modulation to wave in ice attenuation, currently not included in the parameterization included in the WCI model, that must cause a periodic increase in wave in ice damping, resulting in the SWH modulation we observe.

In order to quantify our hypothesis, we estimate a proxy for the average attenuation coefficient applicable on waves that have propagated through the MIZ up to the position of the BOI 19648. For this, we can compute a simple estimate using the ERA5 spectrum value $E_{out}$ outside the sea ice for the peak open water incoming wave frequency $f = 1/12$Hz, and compare it to the energy content at the same frequency at the location of the buoys as reported by the instruments inside the sea ice $E_{in}$. We can then consider the distance of effective sea ice propagation $\Delta x$ to be equal to the depth of the MIZ having a sea ice concentration of at least 0.5 according to the AMSR2 dataset. After correcting for the propagation time lag $\Delta t$ by matching the peak of the incoming wave energy to the peak of the SWH envelope at the buoys, we can estimate the effective bulk

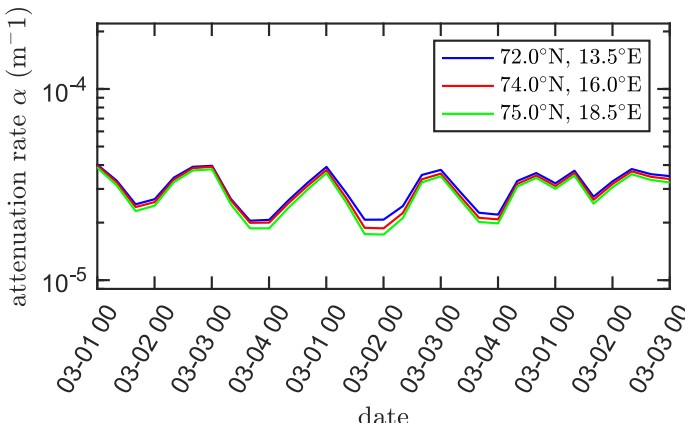

**Figure 10.** Estimates of the bulk waves in ice attenuation coefficient $\alpha$ for $T = 12$ s period waves across the MIZ for the BOI with ID 19648. The distance between the ice edge and the buoy was estimated from AMSR2 sea ice concentration. Incoming wave energy $E_{in}$ was estimated using the ERA5 reanalysis data. Three different coordinates were used as an ERA5 input, for controlling the insensitivity of the results to changes in the exact choice of the ERA5 data selection location (see figure legend for the exact locations). This confirms that our bulk attenuation coefficient proxy estimate is robust.

attenuation coefficient $\alpha$ (with unit m$^{-1}$) as $\alpha = -ln[E_{in}/E_{out}]/\Delta x$. Note that this estimate is only a coarse proxy to be used to estimate typical orders of magnitude involved, and that it neglects several important effects, e.g. the frequency dependence of the attenuation rate and the associated changes in the mean wave period. Moreover, we do not know for sure if the wave in ice attenuation modulation responsible for the SWH modulation we observe occurs in the whole sea ice length through which the waves propagate or if the modulation of the attenuation is concentrated in a smaller sea ice area. Therefore, this approach provides a lower bound for the amount of waves in ice damping modulation necessary to reproduce the SWH modulation observed, in the case where the waves in ice damping modulation is distributed over as long a distance as physically possible. If only a smaller length of the wave propagation track in the ice is responsible for the SWH modulation we observe, this can result in a significantly higher local wave in ice attenuation modulation.

The results are presented in Fig. 10. As visible there, the lower bulk attenuation coefficient $\alpha$ over the whole length of the wave propagation in the ice has a typical value of $2.10^{-5}$m$^{-1}$, respectively $4.10^{-5}$m$^{-1}$, during high, respectively, low, SWH events at the BOI 19648. In other words, the modulation of attenuation corresponds to a doubling of the effective wave attenuation rate averaged over the whole length of the wave propagation in the ice. We note that these values of the attenuation coefficient are typically within the range of values previously observed: Voermans et al. (2021) reports that typical values for $\alpha$ are within the range $1.10^{-5}$m$^{-1}$ from broken pack ice (Kohout and Williams, 2013; Thomson et al., 2018), to $2.10^{-4}$m$^{-1}$ for solid landfast ice, for the typical frequency range considered. Similarly, Wahlgren et al. (2023) reports a typical swell attenuation rate of $4.10^{-6}$m$^{-1}$ to $7.10^{-5}$m$^{-1}$ in the Antarctic spring MIZ.

Unfortunately, calculating the detailed spectral attenuation or the attenuation between buoys is not realistic in the current case study. In fact, the array of buoys was deployed long before the event considered, and the buoys are very far from being aligned with the wave propagation direction. Given the complexity of the MIZ shape and limit, as well as of the bathymetry in the area, and the absence of buoy measurements directly at the edge of the MIZ, we believe that this would make calculating a (spectral) attenuation rate between the buoys or from the open water to the buoys highly unreliable. Therefore, we limit ourselves to considering the SWH modulation pattern and the robust bulk damping estimate, as we believe that looking at finer grain data would be very error prone and highly sensitive to additional factors. Still, our results are a good illustration of the fact that even a relatively moderate modulation of the wave attenuation rate $\alpha$, here likely within a factor of 2 as visible in Fig. 10, can lead to a much larger significant wave height modulation deeper in the ice, in our case a factor of 10. This highlights that even relatively subtle changes in ice conditions can have a drastic impact on the amount of wave energy observed deeper in the sea ice.

At this point, we need to review which mechanism(s) can explain the periodic increase in the wave in ice damping we observe, evaluate whether they are plausible explanations for our observations, and look for the possible signatures or side effects of these mechanisms. A review of the existing literature provides a number of candidate mechanisms for wave in ice attenuation. These are: (i) wave scattering due to the hydrodynamic response of the ice floes, (ii) viscoelastic damping in the sea ice, (iii) laminar or turbulent water-ice friction under the ice due to the wave-induced water velocity, (iv) floe-floe interactions including hydrodynamic interaction between adjacent floes, collisions, rafting, ridging, and crushing, and (v) other phenomena, such as overwash.

If the modulation observed is related to (i: scattering phenomena), this would imply that a major quasicyclical change in the floes packing or floes sizes or shapes takes place, so that the response amplitude operator of the floes (which is a function of their individual geometry and relative positions), and the level of wave diffraction in the MIZ and wave refraction at the ice edge drastically changes back and forth over a timescale of 12 hours. Regarding the array effect of multiple scattering caused by the floes arrangement, this may be a possible explanation given that the ice divergence changes strongly over a 12-hour period and exhibits nearly periodic variations correlated with the SWH modulation. This translates into a modulation of the closedness between adjacent floes and their level of packing, which can influence the level of wave scattering (Meylan and Squire, 1994; Bennetts and Squire, 2009). The other explanation, i.e. changes in the floes shape, appears less likely. It is not realistic to expect a major back-and-forth change of ice floe geometry over such a short timescale: while it is possible for floes to break relatively fast under the influence of waves, consolidating large floes through re-freezing in an effective enough way so that previously broken floes start behaving as single floe is a slower process.

For (ii: viscoelastic-induced damping) to be the explanation of the modulation, viscoelastic effects would need to be significant and the viscoelastic properties of the sea ice would need to change with a 12-hour period. This is unlikely for several reasons. First, in the conditions encountered here, where the ice cover is composed of broken ice floes and is nonhomogeneous, floes flexion and viscoelasticity are likely not a dominating mechanism. Moreover, even making the hypothesis that viscoelastic damping could play a role, it is unclear how the viscoelastic properties of the ice could vary with a 12-hour period. Indeed, while it is well established that the mechanical properties of sea ice can be drastically degraded after, for example, tempera-

ture changes and the impact this has on brine content (Ji et al., 2011; Karulina et al., 2019; Marchenko et al., 2013), which can lead to a rapid degradation of the Young modulus of sea ice, this cannot explain the ice recovering its mechanical properties. Moreover, the atmospheric temperature variations during the wave-modulated event are moderate. Therefore, the fact that we observe a periodic change in the effective SWH attenuation, combined with the lack of evidence for large temperature variations in the area during the modulated wave event, makes this explanation unlikely.

Regarding (iii: under ice friction), since the total amount of ice varies relatively little in the area over a 12-hour period, and we would a fortiori not expect that significant melting and re-icing can happen with the 12-hour periodicity we observe, we do not believe that a large change in the water-ice contact area takes place, so that changes to the water-ice area and the implied laminar flow energy dissipation does not appear as a credible explanation. A change in the turbulence level under the ice could modulate the effective eddy viscosity appearing in the equation driving the water-ice stress and the associated energy

dissipation, even if the total ice-water area does not change. Such modulation in the amount of turbulence under the ice can come from several sources, in particular, the propagation of the tidal wave through the area of interest, floe-floe collisions, or changes in the relative ice-water velocity. However, if the level of turbulence was strongly modulated in the area due to some phenomenon external to the sea ice, for example tides, more damping would have been observed from satellite altimeter data also in the absence of sea ice. This, in turn, makes us believe that if turbulence level changes participate in causing the observed

SWH modulation, these are likely introduced due to some ice mechanisms, such as floe-floe interaction and collision. Change in effective roughness of the sea ice-water interface can also modulate the ice-water stress and dissipation, but for this to play a role here, a periodic 12-hour increase and decrease in the sea ice roughness would be needed. This seems little realistic, as icing and melting mechanisms under the sea ice are not expected to present such strong periodicity, and phenomena such as rafting and ridging would lead to an increase, but no subsequent decrease, in the ice roughness - so that this likely cannot

explain for the periodicity we observe. An additional possible mechanism could be that, while the water-ice area of contact and the turbulence level under the ice are not strongly modulated, the degree to which the motion of the ice is horizontally constrained could change with time, following sea ice convergence and divergence. More specifically, when the ice cover, which is constituted of many broken floes of moderate size, is "open" following a period of sea ice divergence, ice floes could be able to move to some degree in the horizontal direction, so that the slip velocity between the wave-induced water motion

and the ice is reduced. By contrast, when the ice is "closed" following a period of sea ice convergence, the horizontal motion of the floes compressed together and acting as a larger ice sheet may be more strongly constrained, and the slip velocity may be higher. This could lead to different boundary conditions determining the shape of the boundary layer under the ice, and influence the damping induced by under ice boundary layers, as previously discussed in Marchenko et al. (2019). This also could be seen as a secondary effect of floe-floe interaction, since it is the close contact between floes that would prevent the

ice to follow the waves. However, the sea ice concentration never reaches 100% in our models in the area of interest, and SAR images confirm that leads and water openings seem to always be observed around the BOIs, and that floes have moderate sizes compared to the typical wavelength. Therefore, the hypothesis that the sea ice may get so tightly packed that ice floe motion get strongly constrained in the horizontal direction at some times compared to others seems unlikely.

The mechanism (iv: floe-floe interaction) can influence the wave in ice damping in several ways. First, collisions per se can dissipate energy, through both sea ice crushing, inelastic collisions, and hydrodynamic pumping effect between colliding floes (Rabault et al., 2019; Løken et al., 2022; Noyce et al., 2023). Second, collisions, through the hydrodynamic pumping effect and the water jets this creates, inject significant amounts of turbulence under the sea ice (Rabault et al., 2019). This can possibly modulate the amount of turbulent kinetic energy and the effective turbulent eddy viscosity, and create an increase of the damping from the mechanism (iii) previously discussed. The question of the existence and importance of collisions in the MIZ has been debated in the last few years, and our feeling is that the community is currently divided on this topic. This assessment is based on personal communications and discussions received by some of the authors and reviews received during the publication process of Løken et al. (2022); Dreyer et al. (2024). In particular, while some works conclude that floes can slide on top of waves, which combined with the existence of inertia and added mass, suggests that collisions may happen (Shen et al., 1987; her, 2018; Smith and Thomson, 2020), we have also received arguments during previous review processes that floes mostly follow the waves in synchronization, so that there are no collisions actually happening between the floes even if all floes move. However, this last argument disregards several aspects of waves in ice and the MIZ. First, floes are not identical: as reported by the width of the floe size distribution, different floes have different size and shape, leading to different response amplitude operator characteristics in waves, which would imply the existence of collisions. Second, even though waves in ice at any depth in the MIZ are usually dominated by long-crested swells, there is still stochasticity in the amplitude and wavelength of individual waves. Therefore, even two identical floes next to each other, with the same RAO, can experience collisions if two consecutive waves have a height and wavelength different enough. Finally, sea ice cover has many features, such as ridges and keels, that can induce relative motion between floes and create collisions. More generally, we believe that the rarely reporting of floe-floe interactions and collisions may arise, at least partially, from a selection bias: to be allowed to deploy, operate, and recover SD-card-containing instruments that can measure such events, sea ice conditions must be calm enough to allow safe work on sea ice. This is typically not the case for conditions in which sea ice breakup and collisions may occur, therefore, potentially limiting the amount of such events in timeseries records available for analysis. We also note that collisions are the most extreme realization of the floe-floe interaction. In particular, it may be possible to have significant floe-floe interaction, including the creation of water pumping and water jets between floes, even without a full collision to take place. These effects will take place as soon as a significant change in the short-range floe-floe distance takes place, even in the absence of collisions.

We also observe that our findings are roughly similar to what has been reported by Løken et al. (2022): in the corresponding work, collisions and water pumping increasing the turbulent kinetic energy dissipation under the ice floes were found to dissipate around 40% of the total energy input. If we would assume that the dominating factor in the change of $\alpha$ is due to the switching on / off of floe-floe interactions, collisions, and enhanced turbulent dissipation, the doubling of $\alpha$ between low and high SWH conditions would similarly correspond to up to 50% of the energy being dissipated by these mechanisms in the cases when $\alpha$ is highest. Naturally, these are only proxy estimates, and the exact numbers would be slightly modulated by a number of factors, including the effect of wave-current interaction, which, while it was found too weak to explain for the modulation observations, does contribute to the exact values obtained.

Finally, it is possible that (v) other mechanisms may play a role in the present observations. One such mechanism could be overwash. However, there are no drastic changes in the incoming wave field, so it is unclear how overwash levels, which one could expect to be related to the incoming wave steepness, may drastically change. More generally, while one cannot rule out that other mechanisms could play a role, these are, if such is the case, not easily identifiable by the authors and not widely discussed in the literature.

Therefore, the results obtained in the Data Analysis section do not seem to be compatible with explanations (ii) or (v), and explanation (iii) can be a contributing factor but could be present as a biproduct of (iv). As a consequence, this lets us with the possibility that explanations (i) and / or (iv) (possibly in combination to its effect on modulating (iii)) are the most likely phenomena that can explain for our in situ observations. However, this conclusion is obtained by a combination of elimination of other possible explanations and high-level discussion of the mechanisms at play. Therefore, more in situ data will be needed in order to bring a direct positive proof of our circumstanced hypothesis that either a strong change in the scattering properties of the array of floes, or floe-floe interaction, is the explanation for our SWH modulation observation. Although this is not attainable for this specific event from 2021, we can suggest additions to the functionality of future OMBs and similar buoys that would allow us to provide positive evidence of our conjecture. In particular, it should be easily possible to measure floe-floe collisions, since the signature of such events in the acceleration timeseries recorded by instruments on the sea ice is very characteristic (Løken et al., 2022; Dreyer et al., 2024). As a consequence, we believe that extending the OMB firmware with on-board processing for collision detection to quantify the occurrence of events similar to what is described in Dreyer et al. (2024), and transmitting this information together with the wave spectrum as additional data fields is a natural next step that could help to distinguish between explanations (i) and (iv). This should be easily implementable, since the full time series of the sea ice acceleration is already available on the OMB, as it is used to calculate the wave spectrum. It should, therefore, be easy to include a high-pass filtering analysis of the signal to the firmware, and to use this to count collision events and measure their strength relative to the smoother wave-induced signal. We intend to perform such firmware extensions and include these in our buoys in the near future.

The present observations are likely related, as we have discussed, to the convergence and divergence of the sea ice induced by the strong tidal currents in the area and the effect this has on the wave in ice attenuation. This makes the present area particularly interesting to study further, as such tide-driven dynamics should be observed reliably in a relatively wide area where strong tidal motions are present. However, the applicability of these observations is likely more general. In fact, we expect strong sea ice convergence and divergence effects to also occur without strong driving tidal currents. For example, sudden wind events trigger inertial oscillations, which, when they hit the MIZ, will interact with the gradient in sea ice concentration and cause sea ice to close and open. This implies that the dynamics we observe may play a role in storm conditions anywhere in the MIZ, both in the Arctic and Antarctic, though such events may be less reproducible and less regular than in the present tide-driven conditions. Such modulated waves in ice amplitude events have possibly been observed previously in some satellite observations of the MIZ following the passing of storms (internal correspondence and communication), and should also deserve attention in the future, independently of location and the presence of tidal currents. We also show in Appendix C that similar modulation events can be observed in the MIZ in at least another dataset, which is acquired at a different geographical location and season.

A natural question arising after this study is why such modulation has not been reported in the past. In particular, if our most likely explanation that the modulation is ultimately caused by the effect of currents and tides on the sea ice concentration and closedness is correct, similar features could be observed in other contexts. We believe that there can be several explanations as to why we are, as far as we know, the first to report such a modulation. i) Waves in ice data are still relatively scarce and limited in volume. A modulation sufficiently pronounced to attract the attention of someone browsing through their data may rely on a combination of broken sea ice conditions, strong tidal currents, and possibly land masses that provide a fixed boundary condition for the ice motion in part of the domain, or another mechanism that allows currents to cause strong changes in the ice closedness. All in all, observing an 80-90% modulation rate, as we do here and initially attracted our attention, may be quite rare, even in the case if lower levels of modulation may happen relatively commonly. ii) To clearly resolve the modulation, relatively high-frequency sampling of the wave conditions is needed: for example, the modulation in Fig. 3 is significantly easier to see in the OMB-v2021 data (which are sampled every 2 hours), compared to the older OMB-v2018 (which are sampled every 4 hours). Not all buoys may have sampled data at the higher rate of the OMB-v2021, in particular to cut costs and save battery.

However, once we first know that such modulations may be present in the data and actively look for them, it seems to us that these may actually be quite common. Although the point of the present (already long) paper is not to do meta-studies of when such modulations are observed, we can report that a quick browsing through, e.g., OMB-v2021 data collected north west of Svalbard (i.e., a different location) during summer 2022 (i.e., a different year and season), present similar features, as seen in Fig. 15 in Appendix C. A more systematic look at the occurrence of such modulation and an investigation of whether other mechanisms (for example, the impact of currents or wind or inertial oscillation acting on a MIZ presenting a gradient of sea ice concentration) can cause similar features in other datasets would be an exciting direction for future studies. We conjecture that such a study could possibly allow one to indirectly gain more understanding about the key physical mechanisms that play a role in waves in ice attenuation.

## 5  Conclusions

We report direct in situ observations from a set of wave buoys in the Arctic MIZ that contain strongly modulated waves-in-ice significant wave height (SWH) records, with a 12-hour SWH modulation period. The modulation ratio observed between consecutive SWH maxima and minima goes up to 90%, while the associated bulk wave attenuation rate is modulated by a factor of around 2. These data are a good illustration of the fact that, owing to the exponential attenuation of waves in ice with propagation distance, a moderate modulation in the wave in ice attenuation rate caused by subtle changes in, e.g., sea ice conditions, can result in a large change in the wave energy present deeper into the ice.

We show, through a combination of analyses including numerical models, ray tracing analysis, and direct satellite observations in the same area in ice-free conditions, that while tidal modulation takes place in the area, the observed modulation probably cannot be explained by mechanisms such as wave-current interaction, ice drift, or changes of effective wave propagation distance in the sea ice alone. More specifically, our estimations for the strength of wave-current and ice-independent

SWH modulation, either these are based on numerical models or satellite data analysis in ice-free conditions, conclude that a modulation of typically around 30% at most can be expected due to the tidal wave-current interaction in the absence of sea ice. Moreover, numerical models including the sea ice effect indicate that the commonly used wave in ice attenuation parameterizations can explain the typical bulk attenuation coefficient and the envelope of the SWH observed in the sea ice, but that these cannot explain the SWH modulation we observe. These results, though they are obtained from numerical models which contain sources of uncertainty and are not an absolute truth, suggest that some additional mechanisms are playing a role in the observations we report.

Therefore, we deduce by elimination that the mechanism(s) explaining the modulation we observe is likely to be related to sea ice and, more specifically, to a modulation of the waves in ice attenuation coefficient. For this, we hypothesize that a time-varying contribution in the wave attenuation is likely to play a key role in the MIZ area we observe.

Moreover, we observe that the modulation in the SWH is strongly correlated with the local state of sea ice convergence and divergence. Therefore, we conjecture that the missing variability in the waves in ice attenuation coefficient is possibly arising from the effect of either scattering or floe-floe interaction, which intensity can be modulated by the strong state of convergence and divergence in the sea ice observed in the area and the resulting change in ice closedness, as visible in both satellite data, buoys trajectories, and model predictions. In particular, we hypothesize that sea ice convergence and divergence, by affecting the closeness between neighboring floes, may either modify the scattering properties of floe arrays, and / or strongly modulate the intensity of floe-floe interactions and floe-floe collisions by changing the closedness of the floes field. This second hypothesis is consistent with other recent observations obtained from in situ instruments, where the full timeseries of the ice motion were recovered, which analysis indicates that collisions can take place in the MIZ under certain sea ice and wave conditions. This conjecture is also consistent with recent idealized field experiments, in which collisions and the associated forced water motion and turbulence were reported to be able to contribute significantly to the overall wave energy dissipation.

However, we acknowledge that our interpretation of the physical cause for the wave in ice damping modulation is, at present, speculative to some degree, and should be considered as a circumstanced conjecture. In order to firmly confirm our conjecture regarding floe-floe interaction, indisputable in situ evidence of floe-floe interaction dynamics or other modulated waves in ice attenuation mechanism should be collected through a well-defined metrics, alongside collocated observations of wave in ice attenuation across the MIZ extent, and the correlation between both phenomena should be examined. Although this cannot be achieved in the data from 2021 that we report, we suggest that this information can be obtained in the future by programming waves in ice buoys to report statistics about collisions happening in sea ice, in addition to the information traditionally transmitted. We argue that such additional information should be easy to add to the firmware running on, e.g. the OpenMetBuoys, by applying a simple firmware upgrade to buoys to be deployed in the future. Providing direct positive evidence of the existence and importance of such a mechanism would be a major contribution to our understanding of what mechanisms play a role in determining the variability of the observed wave in ice attenuation, and will be an objective for our future measurement campaigns. Testing for the impact of a modulated level of scattering could also be done, either by

660 elimination if a floe-floe interaction metrics is added to future buoys, or by obtaining timeseries from a compact array of buoys and performing a careful measurement of the level of scattered energy.

We also conclude that the area considered, South-East from Svalbard, is very well suited to investigating complex waves in ice and waves-current interaction phenomena. Indeed, sea ice has historically been very reliably observed in the area during the late Arctic winter, and complex sea ice dynamics in the area are forced by strong tidal currents in a reliable and reproducible

way that seems well captured by state-of-the-art models. Therefore, we believe that this area deserves further investigation and that it is well suited for a range of measurements, from using large numbers of low-cost OMB-like buoys in open water and sea ice to, e.g., long-term deployment of bottom-mounted acoustic Doppler current profiler (ADCP) and pressure sensors, which should be easy to deploy and recover due to the limited water depth (typically 35-50m) in the area. We expect possible future findings and improved physical modeling developed based on such measurements to be applicable also in other situations in the

MIZ where the sea ice convergence and divergence are due not to tides, but to, e.g., passing storms and inertial oscillations that lead to a sudden series of convergences and extensions of the MIZ. In particular, we report the observation of similar-looking modulation events in other datasets, that consider waves in ice at another location and season. This suggests that the present modulation event may be a relatively general phenomenon and not purely a one-off curiosity.

*Code and data availability.* All buoy observation data used in the present study are already released as open source data following the

675 publication of Rabault et al. (2023), see the data available on the Arctic Data Center at https://adc.met.no/datasets/10.21343/azky-0x44, or on Github at: https://github.com/jerabaul29/data_release_sea_ice_drift_waves_in_ice_marginal_ice_zone_2022.

Some key codes and scripts used in the present manuscript are available at: https://github.com/jerabaul29/article_data_modulated_attenuation_ waves_in_ice_2021_03 (will be released upon acceptance of the manuscript for publication in the peer reviewed literature). Please use the issue tracker there for any further queries.

SAR satellite data can be seen on Ocean Virtual Laboratory (OVL) viewer, see, for example, the view of the corresponding area available at https://odl.bzh/c840-gF3.

Satellite wave measurements can be recovered using the wavy Python package: https://github.com/bohlinger/wavy. Ray tracing analysis can be reproduced using the ocean wave tracing Python package: https://github.com/hevgyrt/ocean_wave_tracing.

## Appendix A: wave ray analysis

In this appendix, we investigate how much wave-current modulation can be expected following the influence of tides and tidal currents and their interaction with bathymetry trapping effects, as an order of magnitude. Although not as advanced as the full-feature spectral wave model used above, wave ray analysis provides good phenomenological understanding of the wave propagation under the influence of bathymetry and currents, and this can be used as a way to cross-check results and as a consistency check of more complex models. To do so, we leverage simplified models, including the wave action and wave

ray models. These approximate models should not be taken as a ground truth, as they make many simplifications and approx-

imations, and neglect important phenomena including, e.g., nonlinear interaction, which are included in the more advanced spectral wave models. Despite these limitations, they are useful for providing a high-level view of the main phenomena.

Wave-field modulations due to currents are a combination of local and non-local effects. For example, the conservation of wave action on quasi-stationary currents causes an increase in SWH on countercurrents that can be considered and computed as a local modulation (Longuet-Higgins and Stewart, 1964). However, in situ observations also include the signal of non-local modulations, or simply cumulative effects, since waves are constantly interacting with local currents along their propagation path (Masson, 1996; Vincent, 1979; Saetra et al., 2021). One such non-local effect is current-induced refraction, which is considered to be the dominant cause in creating spatially dependent wave height variability at scales of kilometers (Quilfen and Chapron, 2019; Bôas et al., 2020). These effects are linked to wave kinematics, which is incorporated in spectral wave models. However, simplified theoretical considerations are helpful in determining their relative contributions (Holthuijsen and Tolman, 1991).

When considering the effect of local wave action on SWH, the deep-water modulation in wave amplitude $a$ for a steady current having the same (resp. opposite) direction to the wave propagation direction can be expressed as (Phillips, 1977)

$$\frac{a}{a_0} = \frac{c_0}{\sqrt{c^2 \left(1 \pm \frac{2U}{c}\right)}}, \tag{6}$$

where $c$ is the wave phase velocity, $U$ is the current speed, the subscript $_0$ denotes the wave amplitude and celerity value when $U = 0$, and the value $\pm$ corresponds to current having the same, resp. opposing direction as the waves. Here, the wave celerities $c_0$ and $c$ are quadratically related through the Doppler shift equation (Phillips, 1977): $c/c_0 = (1 + \sqrt{1 \pm 4U/c_0})/2$. In our case, the current field has a strong tidal signal and the maximum current speeds in the vicinity of the BOIs are typically about 0.4 m s$^{-1}$ (this value can be extracted either from the Barents2.5 model output [not shown], or from pyTMD and the Arc2kmTM model; see Fig. 14, noting that slightly stronger tides can be obtained at some other locations in the area of interest than what is presented there). The energy modulation according to Eq. (6) is sensitive to wave celerity, and thus wavenumber, such that longer waves are less modulated than shorter waves for a given current speed. Consequently, the modulation of the amplitude is as in Eq. (6) for waves with initial periods $T = 8$ s and $T = 12$ s, opposing (or following, with a sign change including the Doppler effect on $c$) a $U = 0.4$ m s$^{-1}$ current from rest, is about $\pm 7\%$ and $\pm 5\%$, respectively.

To assess the current-induced modulation due to refraction along the propagation path of the waves, we perform a ray tracing analysis using the solver of Halsne et al. (2023). Currents are taken from the Barents 2.5km ROMS model mentioned above and are fully allowed to vary in time during the wave ray propagation. The propagation paths are found to be very sensitive to their initial direction and location, which is a well-known fact for this kind of simplified model (Smit and Janssen, 2019). Therefore, we establish a representative ensemble of wave rays by perturbing the initial direction $\theta_0$ both in the relative positive and negative angle directions, following the typical amount of directional spread present in the incoming wave spectrum. Furthermore, we also included the wave periods $T \in [10, 11, 12, 13, 14]$ s, to include, similarly, the frequency spread of the incoming waves. Consequently, the ensemble was created to be representative of the WAM directional wave spectrum and consistent with what is presented in Fig. 8. The complete ensemble consisted of 40 (initial propagation times) × 5 (ensemble

representing wave directional spread) × 5 (ensemble representing wave periods spread) = 1000 wave ray realizations. A subset of the results are shown in Fig. 11 for illustration purposes. It is clear that the longest waves are sensitive to bathymetry, which can act locally as a wave guide and trap these. In such cases, the ambient current may increase the spreading of the wave rays such that additional rays get caught by the bathymetry. However, wave rays do not always spread more due to the tidal currents, since current-induced refraction depends on the vorticity of the ambient current along the wave ray propagation direction (Kenyon, 1971), which changes in time along the wave ray track.

The wave ray density is representative of the amount of wave energy (i.e., the square of the SWH) that is obtained in a given area at the corresponding time. Therefore, the wave-ray density can be used to derive a proxy estimate for the SWH modulation obtained due to the wave-current-bathymetry interaction. More specifically, we perform a normalized wave ray density analysis for a 12.5 km × 12.5 km region along the track of BOI 19648 (upper middle panel Fig. 11). Here, we count the number of rays within the region and normalize it on the number of rays within the same region, obtained in an additional baseline case without currents (i.e., the baseline case only takes depth-induced refraction into account). As a consequence, this quantifies the amount of modulation introduced in the corresponding region. The square root of the relative density of rays ($\mathrm{RDsr_r}$) is computed as:

$$\mathrm{RDsr_r} = \sqrt{\frac{\mathrm{RD_c}}{\mathrm{RD_{nc}}}}, \tag{7}$$

where subscripts r, c and nc denote quantities that are considered relative, with current, and without current, respectively. We take the square root since the wave action density propagates along the rays and is proportional to the square of the wave amplitude $a$, while the observations correspond to a measure of the SWH, which is linearly related to the wave amplitude.

The time series evolution in $\mathrm{RDsr_r}$ averaged over all ensemble members for the region of interest is shown in Fig. 12c . As mentioned, there are large fluctuations in the single-ray tracing realizations (Fig. 12b, d). These fluctuations are filtered out by averaging the directional and frequency spread representative of the incoming wave conditions (Fig. 12a). The average over all the directions and wave periods shows a clear tidal modulation, which translates into a SWH modulation of around 20–25 % (Fig. 12c). Therefore, we expect refraction to play an important role in the wave-field modulation. The analysis presented here does not take into account important dynamics like nonlinear wave-wave interactions and wave attenuation, and is, as a consequence, only a typical order-of-magnitude estimate. Moreover, the ray theory approximation is known to fail at focal points and at caustics, in which case, wave phases and local interference patterns need to be considered (Smit and Janssen, 2013). However, such patterns are usually observed in very localized areas (typically covering a spatial extent just a few wavelengths), so that this explanation is incompatible with the fact that similar modulation is observed at the 3 BOIs separated by several tens of kilometers (see Fig. 4 for an overview of the distances in km between the 3 BOIs).

Therefore, our analysis suggests that we can expect a tidal signal in the modulations, which is qualitatively in agreement with the modulations in the swell 1 partition from the spectral wave model (see Fig. 9). However, the modulation level expected due to the wave-current interaction is by far not enough to explain the observations by the BOIs presented in Fig. 3. This further confirms the results obtained with the full spectral wave model.

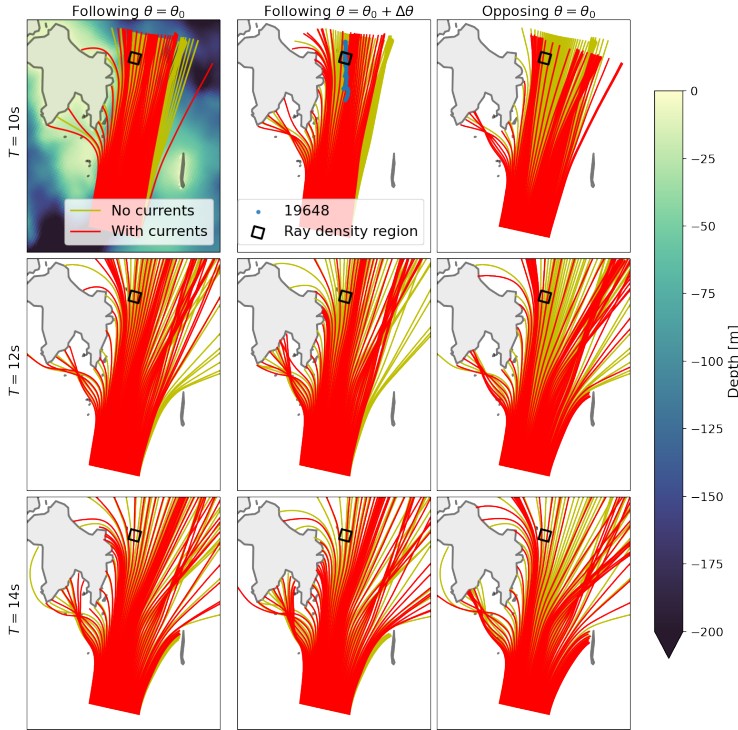

**Figure 11.** An illustrative subset of the 1000 ensemble wave ray simulations performed. Results are shown with (red) and without (yellow, used as a baseline) current forcing. Rows from top to bottom show the ray paths for 10s, 12s, and 14s period waves, respectively. Columns from left to right show (i) wave rays corresponding to the average wave direction ($\theta_0$) for an initial propagation time corresponding to waves following the currents, (ii) wave rays corresponding to one of the perturbed wave directions ($\theta_0 + \Delta\theta$, where $\Delta\theta$ is a deviation from the average wave direction) for an initial propagation time corresponding to waves following the currents, (iii) wave rays corresponding to the average wave direction ($\theta_0$) for an initial propagation time corresponding to waves opposing the currents. Note that the currents evolve in time as the wave rays propagate. The track of BOI 19648 is shown in the upper middle panel, together with the black square indicating the region for the wave ray density analysis. As visible across the figure, the exact refraction of the waves is sensitive to both the current conditions, the wave period, and the exact wave direction. As a consequence, while individual ensemble members may give strong modulation effects, their averaging partially cancels out (see further discussion and analysis of this aspect in Fig. 12).

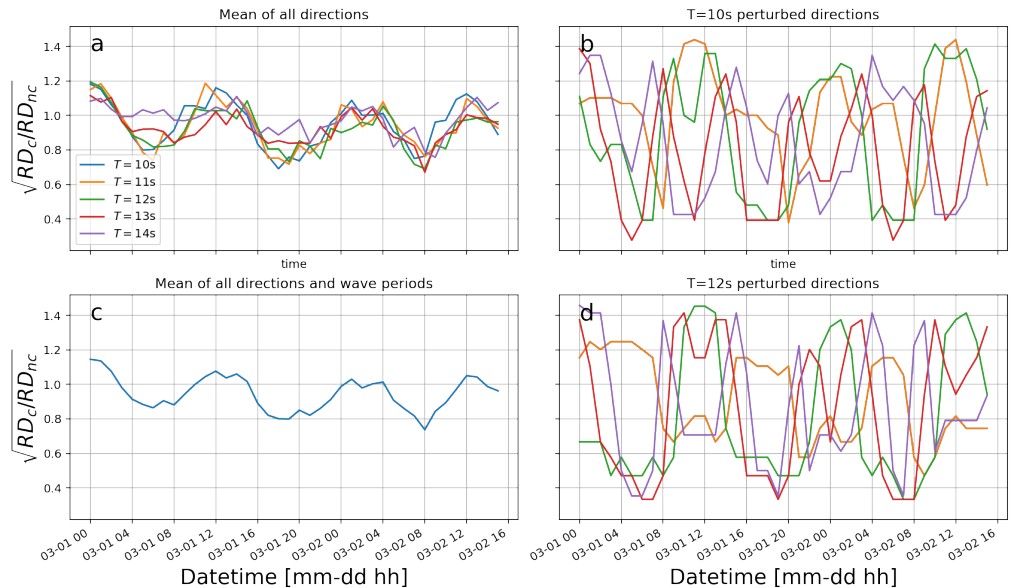

**Figure 12.** Estimates of the expected SWH from wave ray density analysis. Panel a) shows the expected SWH modulation at the location of the BOI 19648, averaged over directions for each frequency used in the rays ensemble, respective to the baseline case without currents. Panels b) and d) show wave ray analysis predictions at the location of the BOI 19648, obtained by perturbing the propagation direction at a fixed frequency. This illustrates the sensitivity of the wave ray dynamics to the ensemble member parameters. Panel c) shows the mean normalized ray density $RDsq_r$ prediction following the track of BOI 19648, obtained by averaging over the ensemble respectively to both frequencies and directions, which is our best estimate for the expected SWH modulation following the wave ray analysis.

## Appendix B: Historic open water satellite measurements in the area, and comparison with tides

In this appendix, we study open water data from the same area to determine whether wave-current or tide-current interaction without the presence of the ice can be observed.

**Retrieval of satellite measurements over ice-free water**

We use the open-source tool wavy[1] to retrieve satellite measurements of the SWH when there is no sea ice in our region of interest. A consistent and homogeneous dataset over many years should be the best starting point when trying to determine whether modulations of significant wave height due to tides and currents are taking place in the area, independently of the effect of sea ice. The Sea State CCI dataset v1 (Dodet et al., 2020) (CCIv1) is used for this purpose. CCIv1 consists of data from 10 satellite missions and is free of discontinuities in the data due to changes in processing or satellite mission. More specifically, we used level 3 processing from CCIv1 where only valid and good quality measurements from all altimeters were retained. We chose the time period 1992 to 2018 and use the dataset as is without any further along-track processing, as e.g.

---

[1]github.com/bohlinger/wavy

in Bohlinger et al. (2019). This allows us to gather a large dataset of significant wave height satellite observations in the area when there is no sea ice present. The corresponding dataset is used for the analysis presented in section 3.3.

## 5.1 Calculation of tidal currents at specific positions

Timeseries for local tidal information at any specific location in the domain are generated using the pyTMD python package (Sutterley, 2023; Sutterley et al., 2019) to leverage modal information from the Arc2kmTM Arctic tide model (Howard and Padman, 2021). The Arc2kmTM is a forward barotropic tide model that includes precomputed maps for the tidal elevation and currents and resolves the influence of 8 principal tidal constituents (4 semidiurnal (M2, S2, K2, N2) and 4 diurnal (K1, O1, P1, Q1)) on a 2km X 2km uniform idealized polar stereographic grid. The area is subject to strong tides due to the features of the local bathymetry, as documented in a number of previous studies that have taken place in the region (Marchenko et al., 2021b; Turnbull and Marchenko, 2022; Kowalik and Marchenko, 2023).

**Correlation analysis between open water SWH from satellite measurements and tidal currents**

In order to further investigate whether the observed SWH modulation could arise from the effect of wave-current and wave-tide interaction independently of the presence of the sea ice, we looked into the direct satellite altimeter observations of the significant wave height in ice-free conditions gathered by the wavy package, and we compared these with the corresponding tide information generated from the pyTMD package and Arc2kmTM model described above. The underlying idea is that if the modulation we observe is due to the wave-current or wave-tide interaction per se and is independent of the presence of sea ice, then it should also be observed by the satellites measuring the local SWH in the absence of sea ice.

Ideally, we would like to gather a dataset of N pairs of SWH observations, $(\text{SWH}_+, \text{SWH}_-)_i, i = 1..N$, with both observations in a pair being obtained with a 6-hours time difference for a variety of different tide phases, so that the SWH at the tidal phase corresponding to maximum and minimum of the SWH following wave-current interaction can be obtained. One could, from such a dataset, compute the average modulation ratio $R = \left\langle \frac{\text{SWH}_{+,i} - \text{SWH}_{-,i}}{\text{SWH}_{+,i}} \right\rangle_i$, where $\langle \cdot \rangle_i$ means "average over sample pairs $i$", $+$ and $-$ refer to the time when maximum and minimum SWH are obtained due to the wave-tidal currents interaction, and $R$ is the SWH modulation ratio due to tidal currents (which can be turned into a percentage by multiplying by 100 for convenience). To make sure that similar wave-current conditions are performed, one could, if enough such pairs are available, limit such an analysis to wave conditions that present similarities in spectrum direction and frequencies with the current case. Unfortunately, since there are no long-term in situ buoy data available at the location of interest, and satellite observations we know of in this area have a repeat rate significantly slower than the 6 hours that this direct approach would request (and a 6 hour repeat rate would be the absolute lowest bound: in order to gather 6-hour pairs of observations covering different tidal phases, an even higher repeat rate would actually be needed), this method is not directly applicable.

Therefore, we instead used our dataset of direct satellite observations by performing a linear regression between the SWH measured by the satellites and the tidal currents obtained from the Arc2kmTM tide model. Our goal is to compute an estimate of the typical average intensity of the tidal current-induced SWH modulation that can be expected for the average representative incoming wave spectrum and SWH conditions obtained in the area. This is not a perfect approach, since one can expect the

actual modulation observed at any time to be dependent on the incoming wave directional spectrum, including its direction and peak frequency properties, as previously mentioned. However, this can still provide an indication of the typical magnitude of the tide-induced open-water SWH modulation effect to be expected. Moreover, since the location considered is close to the coast, the direction for the incoming waves is mostly limited to incoming from the South South West and the Norwegian sea (as we observe in our dataset), or the South South East and the East Barents sea; other directions do not have the fetch necessary to produce significant swells, due to the close proximity of coastlines and the sea ice edge. This approach also assumes that the SWH modulation effect is, in the first approximation, linear relative to the incoming SWH value so that a linear model captures the averaged variability; we believe that this linearization hypothesis is reasonable in first approximation to get a proxy estimate.

In order to perform this regression analysis and check that it is robust to the details of the choice of the exact collocation area, we select a set of local areas around the position of the buoy with ID 19648 during the event, with different spatial extents for the collocation box width, ranging from 30 to 60 km. Such relatively large boxes (relative to the footprints of the satellite measurements, which are typically on the order of 1-10km depending on the satellite and its embarked instrument) are needed to obtain enough collocations to obtain reasonable statistical averages. The most representative such box, corresponding to a side of 45km, is presented in Fig. 13 (left). The center of the boxes is chosen so that the trajectory of the BOI 19648, which shows the strongest SWH modulation, is well covered. For each of these collocation boxes, we find all measurements from satellite altimeter observations obtained from the wavy package that are within the corresponding areas, resulting in sets of satellite collocation SWH measurements $\{SWH\}_d$, with $d$ the collocation box size within the set $\{30, 35, 40, 45, 50, 60\}$ kms. We then calculated the tidal currents in the middle of the box for each collocation time, using pyTMD and Arc2kmTM model data, resulting in the sets of predicted tidal current values. We make the choice of using the middle of the areas for all tidal calculations both for simplicity and because extracting and computing the tidal modes using pyTMD and Arc2kmTM is computationally relatively expensive if many different spatial locations are considered. Since the boxes are relatively small relative to the typical spatial scale for the tide amplitude and phase variability, the resulting mismatch is small, in particular with respect to tidal elevation and current phase variations within each local area, as illustrated in Fig. 13 (right), which compares the pyTMD tidal prediction at the center of the 45 km box to the tidal prediction obtained at the locations following the displacement in time of BOI 19648.

From these data, we can then compute, for each box side $d$, the linear regression between the tide current estimates from pyTMD and the satellite SWH observations. This linear regression analysis is valid for the statistically averaged ice-free wave conditions in the area:

$$\text{SWH}_{\text{average}} = \text{SWH}_0 + \alpha_{u,v} \text{pyTMD}_{u,v}, \tag{8}$$

where $\text{SWH}_{\text{average}}$ indicates the statistically averaged SWH in the dataset observed for given current conditions, $\alpha_{u,v}$ represents, for the same average SWH conditions, the linear variability of the observed SWH as a function of tide currents and $\text{pyTMD}_{u,v}$ represents the estimates of the tidal current obtained from pyTMD. Either $u$ or $v$, which indicate the tidal currents in the East-

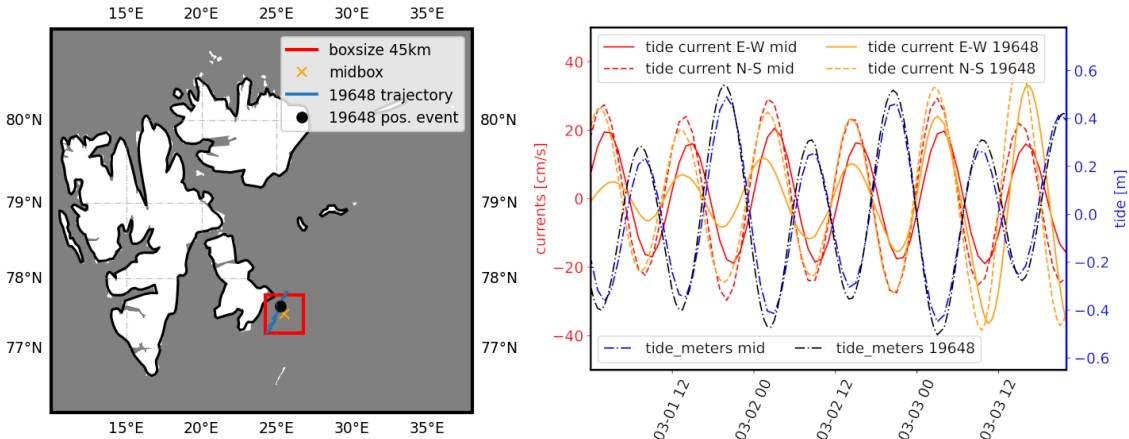

**Figure 13.** Left: illustration of the 45km-width box used to select satellite data SWH measurement collocations. The box contains the trajectory of the BOI 19648, which is deepest into the MIZ and shows the strongest SWH modulation ratio. The position of the instrument 19648 during the most intense modulation event is indicated by the black dot, and is close to the middle of the box which is indicated by the cross. Right: illustration of the tide currents and elevation at the center of the box ("mid" quantities), compared to the ones obtained following the trajectory of the instrument ID 19648 ("19648" quantities). The signals phases and order of magnitude for their obtained amplitudes is similar between the two, indicating that the box extent is correctly chosen. Naturally, the sign of the currents depends on the choice of the referential axis (here, East-West for u and North-South for v), so the apparently opposed phase between tide and current has no particular physical meaning.

West and North-South directions, respectively, can be used to perform the regression. Since, as visible in Fig. 13 (right), the tidal conditions in the area correspond to a propagating wave where the currents and tide elevation are at first approximation in phase and proportionally scaled versions of each other, the difference between both choices remains moderate.

Based on this estimate, the typical average modulation ratio for each box width $d$, obtained for ice-free waters in the statistically averaged wave conditions sampled by the satellite measurements, can be obtained as

$$\text{modulation\_ratio}_{u,v} = \frac{\text{SWH}_{\text{max\_modulation}} - \text{SWH}_{\text{min\_modulation}}}{\text{SWH}_{\text{max\_modulation}}} = 1 - \frac{\text{SWH}_0 + \alpha_{u,v}\min(\text{pyTMD}_{u,v})}{\text{SWH}_0 + \alpha_{u,v}\max(\text{pyTMD}_{u,v})}, \tag{9}$$

where we assume that $\alpha$ is positive (if $\alpha$ is negative, the signs in front of $\alpha$ and the use of min and max should be switched).

The results of the modulation ratio are presented in Table 1. At this point, we want to insist again that these results are statistical averages, obtained based on satellite observations, that present the typical amount of tide-induced SWH modulation for the statistically averaged local wave conditions, based on a linear regression analysis. Moreover, the correlation is performed on the basis of local tidal information, while effects occurring along the wave propagation may also play a role and be out of phase with the local tide and current signal. This means that these results should be taken only as a proxy, or an order-of-magnitude estimate, for the tide-induced SWH modulation effect to expect in the area. Specific cases may result in different

| Box width $d$ (km) | number of satellite collocations | linear regression expected SWH modulation ratio (%), u | lin. reg., v (%) |
|---|---|---|---|
| 30 | 825 | 25 | 28 |
| 35 | 1106 | 24 | 26 |
| 40 | 1451 | 28 | 26 |
| 45 | 1772 | 22 | 27 |
| 50 | 2175 | 24 | 28 |
| 60 | 3091 | 25 | 29 |

**Table 1.** Summary of the linear regression analysis between local tidal currents produced from pyTMD using the Arc2kmTM model, and satellite observations collected using the wavy package of the SWH in a box of width $d$ around the area of interest. Enough collocations are obtained to derive statistically significant modulation ratio values. The modulation ratio values derived are roughly similar when derived from either the u (E-W) or v (N-S) current components of the pyTMD tidal current estimate, which is consistent with the observation from Fig. 13 (right), that the local tide signal is dominated by a propagating wave, with both u, v, and the tide elevation moving in phase. The typical modulation ratio observed is between 20 and 30%. The standard deviation uncertainty on the regression slope, and, therefore, on the SWH tide-driven modulation ratio, is typically within 15% of the estimates obtained (not reported here), which confirms the confidence in the values of the statistically averaged modulation ratios presented. The modulation ratio found shows no strong dependence on the box width, which confirms that the analysis is not overly sensitive to this parameter. The table columns indicate the box width used for each analysis, the number of satellite collocations obtained for the corresponding box width, and the SWH modulation ratios derived using Eqn. 9 based on either the u or v tidal current components, respectively.

values, but the results presented in Table 1 can be expected to give an estimate of the typical modulation intensity expected on average.

Despite these methodological limitations, the values presented in Table 1 are typically consistent with the findings presented earlier in the manuscript, in particular the results from wave ray analysis and from the wave model runs shown in Fig. 9. A typical statistically averaged modulation of 20-30% is obtained from our analysis, which corresponds well to the relative differences found above. This would indicate, similarly to the findings from the previous section, that the wave-tidal current interaction per se is unlikely to explain the modulation we observe from the BOIs. This is also similar to the typical amount of modulation reported in other areas such as, e.g., Gemmrich and Garrett (2012).

We have applied a scaled version of Eqn. 8 to the specific modulation case observed in the BOI 19648 data. By scaling the SWH and the SWH modulation obtained in Table 1 to match the peak SWH obtained for BOI 19648, and computing the tide from pyTMD at the exact positions reported by its GPS track, we can compute the statistically averaged wave-tidal current modulation expected for the SWH reported by BOI 19648. The results are presented in Fig. 14. As visible there and in agreement with the discussion above, the observed SWH modulation is not explained by the statistically averaged tidal current effect. Moreover, the phase of the modulation obtained from the linear regression analysis applied to the predicted tide at the location reported by the GPS track for BOI 19648 is also partially mismatched with the observation of the SWH modulation. Although we acknowledge that our tide current estimates are imperfect as they are obtained from a numerical model with

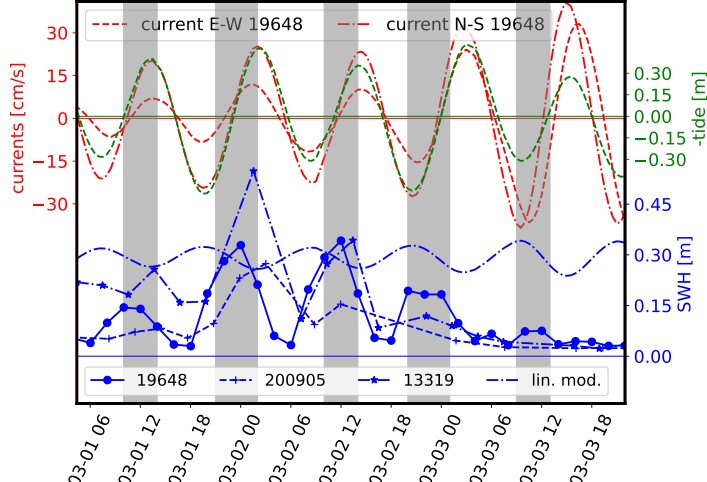

**Figure 14.** Application of the linear regression analysis results to the BOI 19648. The tidal currents and tide elevation following the position of the BOI 19648 (as reported by its GPS track) are computed using pyTMD and Arc2kmTM. Based on this, the linear regression corresponding to Eqn. (8) is scaled linearly to match the peak SWH recorded by the BOI 19648, and the associated statistically averaged modulation in the area is computed (indicated by the "lin. mod" curve). The obtained linear modulation is not able to reproduce the observation reported by the BOI 19648 in the neighborhood of the strongest modulation event (03-01T12 to 03-02T18). This indicates that the modulation observed at the BOI 19648 does not match with the statistically averaged tide-induced modulation observed by performing a linear regression locally in the area, which suggests that another mechanism is possibly the driver for the modulation observed at BOI 19648.

finite resolution, we consider that this is additional evidence that another mechanism is likely to explain the observed SWH modulation. We also observe that the tidal current amplitude at the location following the track of instrument 19648 increases toward the end of the event, which is in agreement with the findings in Fig. 9 that a stronger current-induced modulation is obtained toward the end of the timeseries.

## Appendix C: An example of similar modulation from another OMB-v2021 dataset

Similar looking, though slightly less pronounced, modulation can be observed in other datasets. For example, the OMB-v2021 AWI-UTOKYO-2022 data from the data release paper Rabault et al. (2024) [2] contain a similar modulation, as visible in Fig. 15. The modulations shown here are only the most obvious ones, observed after a quick browsing of the data with the naked eye, and many more events that can correspond to weaker modulations are visible in this and a range of other datasets.

[2]this specific data record is available at: https://github.com/jerabaul29/2024_OpenMetBuoy_data_release_MarginalIceZone_SeaIce_OpenOcean/tree/main/Data/2022_AWI_UTOKYO

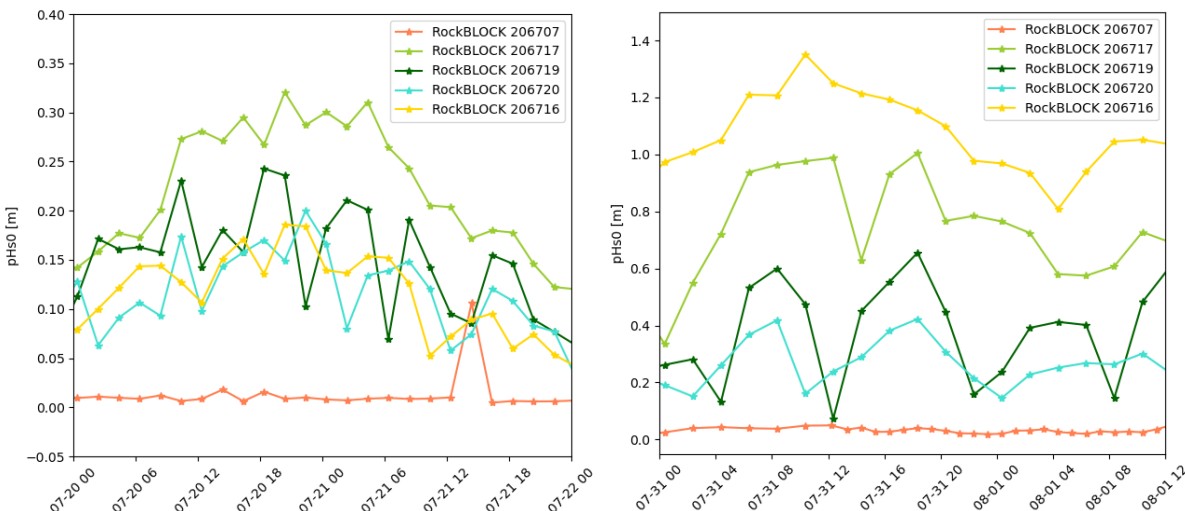

**Figure 15.** Two other examples of waves in ice modulation. These examples are from the summer 2022 (a different year and season), North-West of Svalbard (a different location). While the modulation discussed in the main body of the present paper is particularly strong and well defined, we observe many more similar examples of candidate modulation events in other OMB dataset. $pHS_0$ stands for "processed" (i.e., where the low-frequency noise has been cut as necessary, see Rabault et al. (2022)) significant wave height from the 0th order moment of the PSD.

*Author contributions.* **Conceptualization**: J.R., T.H., A.C., A.C. **Data collection and curation**: J.R., M.M., Ø.B., G.H., F.C., S.H., A.J., G.S., P.B., J.B.D. **Formal analysis and Software**: J.R., T.H., A.C., A.K., J.V., N.H., A.B., A.M. **Funding acquisition**: J.R., Ø.B., K.H.C., A.B., L.A., A.J., A.M., T.W. **Investigation**: J.R., T.H., A.C., A.K., J.V., P.B., N.H., Q.Z., A.B., L.A., L.W.D., K.C. **Methodology**: J.R, T.H, A.C., A.K., J.V., N.H., Q.Z., A.B., C.P., T.W. **Project administration**: M.M., Ø.B., F.C., K.H.C., A.B., J.R., A.M., T.W. **Validation**: J.R., A.C., T.H., T.N., G.H., L.A., G.S. **Visualization**: J.R., T.H., A.C., S.H., F.C. **Writing - original draft preparation**: all. **Writing - review & editing**: all.

*Competing interests.* The authors declare no competing interests.

*Acknowledgements.* We gratefully acknowledge funding from the Research Council of Norway project 276730 (The Nansen Legacy). MM gratefully acknowledges funding from the FOCUS project, funded by the Norwegian Research Council (grant NFR-301450), which also funded the instruments v2021 used in this study. AC and JBD gratefully acknowledge funding from the MakingWaves project from the Norwegian Research Council (grant NFR-325654). We want to thank the crew of the icebreaker Research Vessel Kronprins Haakon for the invaluable help and support deploying the OMBs buoys during the PC-2 winter process cruise in February 2021.

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
