# Peer review of "Buoy measurements of strong waves in ice amplitude modulation: a signature of the impact of sea ice closedness on waves in ice attenuation"

_EGUsphere, 2024_

## Author Comment (AC1)

**Answer to Reviewer 1, Review round 1**

**"Buoy measurements of strong waves in ice amplitude modulation: a signature of complex physics governing waves in ice attenuation"**

**The Cryosphere, egusphere-2024-2619**

This manuscript presents unique observations of ocean wave attenuation in sea ice, in which the attenuation is modulated on a 12-hour cycle. The increased attenuation coincides with sea convergence driven by tidal and/or inertial currents. The observations are clearly documented, including contextual information on sea ice and oceanographic conditions. The manuscript carefully presents and evaluates different hypotheses for the underlying mechanisms, before concluding [cautiously] that the increased attenuation arises from floe-floe interactions during ice convergence.

I find the analysis herein to be rigorous, and I am fully convinced that the observed changes in the waves are related to changes in the sea ice. That said, I have recommendations for major reorganization of the work and reframing of the results. I think the attenuation estimates should be the very first part of the results (and thereby more central to the paper). I think the wave-current analysis and "extra" hypotheses should be evaluated after the sea ice mechanism is evaluated, possibly as appendices or discussion material. It is important to retain this material and show that currents cannot explain the observed modulation, but in the present form the 'reader-fatigue' from this material undermines the impact of the attenuation results.

We want to thank the reviewer for their kind words, and we note that the reviewer is generally positive about our data, results, and manuscript, though the reviewer and us may have different opinions about what is best suited as a methodology / goal for the manuscript, and we may disagree on which focus we want to give to our work.

We agree that the paper is currently quite long and goes into many details. We are fine moving significant parts of the technical discussions into appendixes, and reordering some of the scientific content of the paper. This will be a quite major re-organization of the paper and a significant amount of work, though this does not change the scientific content per se.

For reasons discussed later in this answer, the main point of this manuscript is to demonstrate that the modulation effect likely comes from the effect of the sea ice, and that this is likely the effect of sea ice convergence and divergence, i.e. ice becomes more or less closed. While the reviewer seems to assume given their comments that it is obvious that the

modulation we observe comes from the effect of the ice, we believe that this is not as clear a priori as the reviewer seems to assume: complex currents and bathymetry effects may cause strong, time-dependent, non-local modulations to the wave height. Therefore, our focus is that the main point of this manuscript is to demonstrate that subtle changes in sea ice conditions are the most likely explanation for the modulation observed. We want to keep this focus as it is and not expand it overly, as we believe that it is necessary to carefully validate this affirmation. In particular, we are not willing to extend the manuscript by doing ad-hoc tuning and comparison of waves in ice attenuation models, which would need to be done in a very careful way and with a larger dataset split into some training and validation subsets to avoid HARKing and allow us to offer actual predictive power and scientific value. This can be the focus of future work, which ideally should extend their dataset basis to more cases than just the one we present. Said otherwise, we believe that this research field is already suffering from many studies over-fitting parameterizations with free parameters to individual small-scale field deployments, and we do not want to add to this confusion. Therefore, we want to establish that a physical parameter is important in determining wave in ice attenuation rate, but we believe that to actually develop a parameterization, a much wider body of input data should be considered, which is outside of our scope.

Therefore, we are willing to address most of the small comments made by the reviewer. However, we unfortunately disagree regarding i) the point about the tuning of existing attenuation models (though we are not really sure of what the reviewer had in mind, and if it was a request for adding more work, or just a side comment, see the discussion below), and ii) the suggestion to compute attenuation rates directly from computing ratio of spectra between pairs of buoys. Going into more details, the reasons for this are as follows:

Regarding i), we believe that a relatively "simple" ad hoc and a posteriori tuning of free parameters with time-dependent values (since, as we already show with simple models (WIC runs) that have generally similar characteristics to the complex models suggested, time-independent parameters in wave attenuation models are not able to reproduce the observations) in existing models to fit and reproduce our observations, would be a methodological mistake akin to HARKing. While we agree that we could reproduce our (or, really, any) observation by performing ad-hoc a-posteriori time-varying parameter tuning in existing models, we believe that this would create an "artificial" agreement and actually imply nothing about the true validity of the models and the physics they contain.

Regarding ii), the array of buoys is, by a wide margin, not aligned with the direction of wave propagation. As a consequence, the differences in attenuation are arising from a combination of factors, ranging from depth in the sea ice, to the sea ice area through which each wave ray has propagated. This is a major reason why we put so much emphasis on comparing our buoy observations to models, rather than "simply" computing ratios between buoys. This will be made clearer in the next iteration of the manuscript. We note that this is well aligned with some recent findings and discussions, see for example an enlightening recent study on this topic: https://doi.org/10.3389/fmars.2024.1413116 .

As a result, we want to keep the focus of the manuscript as it is now: showing, by eliminating alternative explanations, and by leveraging detailed model runs with different sets of physics enabled or not, that the modulation observed most likely comes from the effect of the sea ice, and is most likely an indirect consequence of sea ice convergence and divergence.

Given the buoys configuration, proving such a fact is in our opinion not trivial, and this is why this takes most of our efforts in this manuscript. Since the manuscript is already quite long and all our data are openly available, we believe that this focus makes sense in the present context. Naturally, further studies either by us or by other groups can further investigate these data, building on the understanding we generate here.

In the following, we follow The Cryosphere's revision process that, at this stage, only answers to the reviewer are provided (and we do not provide an updated manuscript yet).

Once the attenuation estimates are more central in the paper, the results can be reframed to acknowledge that small changes in attenuation rate make big differences in wave observations over long propagation distances. Thus, the observed factor of 10 modulation in significant wave height (SWH) from 0.03 to 0.33 m arises from a mere factor of 2 modulation in the attenuation rate. In the context of prior waves-in-ice studies, this a very modest change in the attenuation rate. For example, Rogers et al 2016 (DOI:10.1002/2016JC012251) find similar changes that occur simply from differences in the shape and maturity of pancake ice floes. Other studies find that a factor of 2 change in the attenuation rate can occur between the compact edge of the marginal ice zone and the more diffuse interior (Hosekova et al 2020, DOI: 10.1029/2020JC016746). With this in mind, I disagree with the interpretation that the convergence of the sea ice "switches on" a new mechanism related to floe-floe interactions (collisions, etc). Rather, I think that convergence of sea ice causes subtle changes in sea ice concentration and/or thickness (through volume conservation), and that causes an increase in the attenuation rate. The increases might be reasonably well-described by existing parameterizations (Meylan et al, 2018, DOI: 10.1002/2018JC013776; Rogers et al 2018, NRL/MR/7320--18-9786). Those existing parameterizations have tuning parameters that are not tested here, so we cannot say whether new formulations are required.

We agree that we can change the order of the manuscript and make this point about the "relatively moderate" (factor 2) change in the bulk attenuation coefficient early in the manuscript. We are fine adding the references that the reviewer mentions, and discussing that these point to a wide range of bulk damping parameters.

We are a bit unsure about what the reviewers means / wants regarding the tuning parameters that are not being tested. On our side, we do not want to perform ad hoc tuning of existing parameterizations and make this part of the manuscript. We also believe that the nature of the variability we observe here is quite different from the variability observed in the studies that the reviewer lists. There are several reasons for this, as presently highlighted in the manuscript:

- The sea ice conditions in some of the studies that the reviewer points to were very different from the ones we had: our buoys were deployed in close drift ice, and these conditions were definitely not conditions dominated by grease ice and pancakes as in several of the studies cited by the reviewer. Moreover, we observe periodic changes in the SWH and corresponding attenuation rate over 12 hours. As discussed in the manuscript, this cannot reasonably be expected to come from periodic changes in the ice conditions: the sea ice will not periodically change from grease to pancake or other state back and forth within a period of 12 hours.

- We do not have observations at a level of granularity that is fine enough to validate if the attenuation rates periodically doubles / halves over the "global" domain considered everywhere, or if the attenuation rate remains the same in large areas, and does much more than doubling / halving locally. This was our initial motivation for keeping the bulk attenuation rate plot for the discussion part of the manuscript, and not giving it too much focus, though this can be made clearer and will be now discussed in the manuscript explicitly. We will make this clear, by explaining that the damping rate we provide in the current Fig. 14 are bulk rates over (relatively) large distances, but that we do not know the details of short-scale spatial variations: this may correspond to either a general doubling / halving, or to much larger changes on smaller areas. This is hinted at in recent studies, e.g. https://doi.org/10.3389/fmars.2024.1413116 that illustrates how difficult and potentially tricky it can be to relate local and "large-distance-averaged" damping rates. A significantly larger number of buoys, deployed in a denser network, would be necessary to answer these questions with good confidence - we will make this clear in the next version of the manuscript too.
- Regarding the general direction of tuning / testing the tuning of free parameters. This may be a bit of a different discussion / side thread, but we find that it is potentially problematic, from a fundamental methodological and scientific viewpoint, to tune models in an ad hoc, a posteriori way to fit observations, which is why we do not perform such a task here. We agree that, since most waves in ice models have free parameters[1] that can be tuned, and observations are relatively few and scarce, we can tune any of the models presented in the literature to represent any field observations from a few buoys (especially so if we allow dynamic time-dependent tuning of the parameters as a function of some ice high level properties). However, we believe that this i) does not offer any insight into the physics, as any model independently of the physics it implements can be tuned in such a way, ii) holds no predictive power for future studies and operational models, as the tuning has to be done for every set of observations individually, iii) can easily turn out to be a subtle form of overfitting / "HARKing": effectively, by doing so we would hypothesize (tune in an ad hoc way) what the free parameters in the models should be after observations are taken, so that a good result is guaranteed to be obtained independently of the correctness of the input model and its physics. This, in our opinion, gives a false feeling to the reader about the performance of these models and the fact that these focus on the right physics. This is the core reason why we use models here not to reproduce our observations through ad-hoc tuning, but only to highlight that interesting physics are happening and cannot be explained by simple bathymetry or current effects included in such models, nor by standard non-tuned wave in ice attenuation models, so that the features observed deserve attention.
- Moreover, we are a bit skeptical that the physics present in these models can realistically be expected to lead to the periodic variations that are observed. This is already discussed at length in the manuscript (pages 28-29). In particular, we are not
* * *
[1] usually at least 2 or more parameters are available to perform tuning of such models, to the best of our understanding up to 6 for some of the most complex and least-constrained models; this is more than enough to "draw an elephant and make his trunk wiggle": https://www.nature.com/articles/427297a , https://en.wikipedia.org/wiki/Von_Neumann%27s_elephant

convinced that volume conservation leads to thicker ice when the ice is more closed, which can reverse to thinner ice when the ice is more open. We believe that this may be true for a layer of grease ice, but not really for solid ice floes: ridging may happen, but this is not a "reversible" process: the sea ice does not "unridge" when it opens again. This is the main point of the discussion on pages 28-29, that considers a wide range of possible sea ice characteristics changes that are taken into account in the physics of these models, and that we do not believe can be expected to change periodically and be reversible. If we believe that we cannot expect the physics present in these models to be "reversible", we do not believe that applying these models with parameter tuning to match our observations makes sense, following the point discussed above.

So, while we are willing to add a short description about the points above, in particular highlighting that ad-hoc tuning in time of the free parameters in wave in ice damping models could reproduce any attenuation rate observed, including the one we present, we do not believe that this could help advance the scientific discussion and understanding in the present case. This is why we do not perform such an analysis, and this will be made clearer in the next version of the manuscript.

Naturally, we remain open to the idea that a scientist may find a way to integrate more physics and causality into existing models, and come up with an a-priori theory and model that could quantitatively explain, without HARKing, our observations and other observations without the need for ad-hoc tuning. If so, we believe that this would be a significant advance, and we would be delighted to read about such work. Our data are fully open source, so that any member of the community can leverage our data to do such work. But this is beyond the scope of the present work, and would probably require considering a much larger dataset to be convincing: here, we simply focus on showing that complex features are observed, that cannot be a priori explained (without ad-hoc a-posteriori tuning) by existing models.

Specific comments:

The introduction could be a bit more careful not to overstate the ongoing buoy revolution. Certainly, more and more buoys are being developed and deployed (and this is great). Of the nine OMBs deployed for this study, only a few ended up in the analysis. We should humbly remember that works like Doble et 2006 (DOI: 10.3189/172756406781811303) deployed almost as many buoys 20 years ago, with similar capabilities.

We are fine to discuss older deployments and tone done on the low-cost buoy developments, and highlight to some higher degree that this is a continuation of a trend. However, the fact that we have reached 10x cost reductions and fully open source designs means a lot for the ability to scale up measurement volumes in our opinion. For example, the present study would not have been possible before this 10x cost reduction. In particular, as the reviewer points out, we had to deploy 17 buoys in total over the corresponding cruise to "be lucky" and have a particularly interesting signal on 4 buoys - we believe that this is an argument for, and not against, the need for more and larger deployments.

Moreover, as pointed above, we would ideally like to have even significantly more buoys (we are now actually routinely deploying 30-50 buoys per cruise, instead of 17 here). This is made possible by these cost cuts.

Fig 2 shows clear modulation of the wave spectra within the sea ice. The next logical step to compare with prior studies is to calculate the spectral attenuation rates (and then explore how this is modulated on the 12-hr cycle). In present form, that does not occur until page 31, and even then it is only a bulk attenuation rate. A spectral calculation might reveal more physics, including exploring the power laws described by Meylan et al 2018. If the goal of the paper is to show the cyclic convergence of sea ice changes wave attenuation, then please calculate the attenuation!

While we agree in principle with the comments by the reviewer, we do not believe that this is applicable to the present case. As visible in Figs. 6 and 8, the waves are coming from the South-West direction, propagating towards the North-East. The buoys we have in the area are, by and far, not aligned with this direction of propagation. This is one of the key reasons why we put so much effort and emphasis into running full wave models that include bathymetry and current effects both with and without ice, and we spend so much time comparing to the open water conditions. In particular, it would be deceiving to compute an "attenuation rate" between buoys 200913 and e.g. 19648, since the attenuation difference does not really come from the propagation of the waves "from 200913 to 19648", but rather from the differences along propagation of the waves through different areas and depth of sea ice through the MIZ on their way to each buoy individually. This will be made clearer in the text.

As a general note, we believe that this is a common, and often overlooked, weakness of waves-in-ice propagation studies: as buoys are generally either imperfectly or not at all aligned with the wave propagation direction, corrections and projections are usually applied to obtain attenuation rates. This can be complex and a source of errors depending on the case considered and the complexity of the local bathymetry and currents, which can result in complex shape for the wave rays (see e.g. discussions about these aspects in https://doi.org/10.5194/egusphere-2024-2104 , https://doi.org/10.1017/jog.2022.99 , https://doi.org/10.3390/jmse12112036 ). Similarly, the buoy-to-buoy comparison without a full spectral model being run can lead to many subtle artefacts. A particularly good illustration of this fact was recently presented in https://doi.org/10.3389/fmars.2024.1413116 , and this will be highlighted in the next version of the manuscript.

We believe that, as a consequence, attenuation studies should give more importance to running actual full-blown spectral wave models and comparing these to observations, rather than computing ratios of the signal at buoys that may not be aligned with the wave direction of propagation, which can be deceiving. This is in particular true in complex bathymetry and current situations, as the ones that we have at present: see fig. 9 and 11, that show that a significant (5-30% in the most extreme case, though this is significantly less than the 90% modulation observed by our buoys) modulation effect is present due to current effects. Therefore, directly comparing the spectra between buoys at different locations and deriving a fine-grained attenuation rate from this comparison would possibly include large sources of uncertainties and errors and be misleading.

These are the key reasons why we want to limit ourselves to computing and plotting the bulk attenuation rate analysis between open water and one single buoy as we do it now to get a feeling for orders of magnitudes, but we believe that going further in a point-to-point attenuation ratio analysis between two buoys should be avoided.

Similarly to the other comments, if the reviewer disagrees with our view, the data are openly available and more analysis can be performed independently of our work - however we would just not feel comfortable doing so ourselves, and we believe that this would be much more involved and uncertain than just computing an attenuation rate based on comparing the spectra for two of the buoys.

The work to show that currents and other non-ice mechanisms are insufficient to explain observations is very thorough, but it is almost a distraction. It's pretty clear from buoy 200913 that there is no modulation near the ice edge. So it's not the incident wave field that is changing. In particular, the historic/statical open-water wave analysis would be better placed in an appendix.

Our opinion is that the analysis of currents and other non-ice mechanisms is more necessary than what the reviewer believes / highlights in their comment. In particular, it could be possible in theory to have no modulation on the periphery of the domain, but to still have very strong and localized focusing effects due to bathymetry and / or currents closer to the coast, leading to modulation closer to the coast where the inner buoys are located. This is made even more complex by the fact that our buoys are not well aligned with the wave propagation direction. This is why we put so much emphasis on testing the hypothesis that the modulation comes mostly from the effect of the ice, and not from alternative effects. We can make this even clearer, and state explicitly that very strong modulation can be obtained locally due to currents and bathymetry (see, e.g., https://doi.org/10.1016/j.ocemod.2022.102071 , https://doi.org/10.1175/JPO-D-20-0290.1 , https://doi.org/10.1175/JPO-D-23-0051.1 ), so that we need to make sure that this is not what we are observing.

Despite these considerations, we are fine to reduce the discussion in the main body of the text, and put the more technical and lengthy parts in new appendixes.

Bottom of p 27: the statement that the CICE model used as input to the wave-current-ice ("WCI") model results reproduces the "time dynamics of the sea ice cover" is not supported in a quantitative way. Does CICE reproduce the convergence and divergence calculated from the buoys (Figure 3)? More broadly, the tone of this section is "well, the WCI model does not show the modulation, so a new mechanism must be needed". My alternate interpretation is that the WCI model has ice damping parameters that could be tuned for convergence of sea ice (increasing concentration, thickness, or both). Figure 5 shows that sea ice concentration is very high at buoy 19648, but it is not 100%. Convergence could cause it to increase.

We believe that we already show that the CICE model reproduces the ice dynamics. As pointed by the reviewer, this is the point of Fig. 3: the rightmost part of the figure shows both the divergence computed from the buoys of interest following the "triangle element method" (solid red line), and the corresponding divergence computed from the CICE model (dotted red line). As visible there, the correlation is clear "with the naked eye". Naturally, this is a complex case due to bathymetry and currents, so that minor time delays and peaks happen, but overall the general patterns are quite well recognized with the eye. To make this even clearer and more formal, we have computed the Pearson correlation coefficient between the BOI divergence and the model divergence, and the value obtained during the active phase

of the modulation event is 0.71. This is convincing quantitative evidence that the CICE model is able to reproduce the convergence and divergence effect. This will be added to the manuscript.

We agree that it is likely not the sea ice convergence and divergence per se, but its effect on, among others, the sea ice concentration and "closedness" of the ice, and how this influences the physics governing wave attenuation, that may impact the attenuation rate observed. However, the sea ice concentration is not directly measured by the buoys, and, therefore, it is not possible to compare the sea ice concentration and its "closedness" from the CICE model to in-situ data. However, since these are directly constrained by the local level of convergence and divergence, which match well between observations and models, we have a proxy evidence that the CICE model is doing well on these aspects. This will be made clearer in the manuscript.

Regarding the impact of the sea ice concentration on the model: sea ice concentration is already taken into account through the linear weighting of the open water vs. ice terms in the spectral model, see the description of the model setup on page 11. While we agree that this is based on a relatively simple weighting method, this is to the best of our knowledge the standard in the field. We do not think that ad-hoc tuning of other free parameters inside existing wave damping parameterization, without a solid explanation to back it, would be a good practice with as little data as we have here: as discussed above, we could reproduce any behavior with tuning. Moreover, we believe, as pointed above, that developing new models is outside of our scope of the present work and would likely require considering significantly more and more diverse data, as discussed earlier in the answer.

Bottom of p29: the literature is pretty clear that ice floes do not follow the waves in "synchronization" but rather they slide down-slope on the face of the waves and have 'added mass' that introduces phase changes. Thus, they definitely collide. See Shen et al 1987 (DOI: 10.1029/JC092iC07p07085) and also Herman et al (JGR, 2018). Also the Smith and Thomson 2020 stereo work (already cited in the intro).

Thank you for pointing to these references, we agree that they are relevant, we will add a few sentences and a short paragraph discussing these.

Technical corrections:

The lack of line numbers in the PDF is frustrating

We agree with the reviewer, actually we had a discussion on this point with the editor at the submission of the manuscript, see the letter to the Editor. Unfortunately, given how the submission system works, there is nothing more we can do when using the "ArXiv submission method", as far as we know.

The usage of a hyphenated 'waves-in-ice' or simply 'waves in ice' is not very consistent. I suggest the convention of hyphenation when the phrase is used as an adjective (e.g., "waves-in-ice physics are estimated") and no hyphen when used as a noun (e.g., "waves in ice are measured…")

We will go through the manuscript and make this consistent to the best of our abilities. The journal may have its own policies regarding hyphenation, and this will also be handled by the journal "redactors" at a later point when the proofs get generated, if this is considered necessary. We also believe that, while it is true that our command of the English language is not perfect and most of us are not native English speakers, this does not impact the ability to convey scientific ideas.

Top of p3: it is the *gradients* in wave radiation stress that transfers momentum to the ice and water. Without gradients, the radiation stress is simply an ongoing flux of momentum (but no transfer).

Thank you for your comment and in-depth reading of the manuscript, this is indeed not clearly explained at the moment, we will improve on the text.

Top p3: another recent fetch study is Brenner and Horvat (2024): https://doi.org/10.1029/2024JC021629

We are fine to add a couple of sentences discussing this work.

P 11: The inclusion of possible temperature modulation (and associated changes in sea ice rheology) is a good point, but evaluating with ERA5 seems like a poor match to the task. ERA5 would probably only show temperature changes if it also had modulation sea ice, which it does not. Surely the CICE model employed herein has temperatures?

The reason for looking at ERA5 data is that, given the 12 hours period observed in the buoy data for the modulation, only a drastic and fast change in the atmospheric temperature and associated heat fluxes with the sea ice would have any chance to exert a forcing that is strong enough to modify the ice conditions in a way that such a signal could be observed. ERA5 is a well-known and robust dataset that is suited for checking this, and, since this is not happening in ERA5 2m temperature data, there is no need to further consider complex sea ice temperature profile models to rule out this possibility: the forcing that could trigger such drastic changes is not present to start with.

Regarding the exact model runs we performed, these were run with CICE (metroms system) using Arome-Arctic as forcing. There too, there is no sign of large, periodic variations in 2-m temperature in the forcing of the ice model.

---

## Author Comment (AC2)

**Answer to Reviewer 2, Review round 1**

**"Buoy measurements of strong waves in ice amplitude modulation: a signature of complex physics governing waves in ice attenuation"**

**The Cryosphere, egusphere-2024-2619**

The manuscript describes a detailed analysis of an observed wave event in the Arctic MIZ via wave buoy measurements that exhibits a large amplitude modulation over a period of 3 days in the Spring of 2021. The 12-hour modulation period strongly points towards an effect of currents/tides. A wide range of datasets are then used to test this hypothesis. The main finding is that currents and tides alone cannot fully explain the magnitude of the observed modulation and that processes related wave-ice interactions are likely the main cause of this effect. In particular, a periodic switch between ice drift convergent and divergent regimes, which leads to stronger vs weaker ice-induced wave attenuation, respectively, is most likely what explains the modulation. A discussion of the physical mechanisms that can cause wave attenuation in ice-covered seas leads the authors to conclude that an on-off switch of floe-floe interactions, e.g. inelastic collisions and hydrodynamic pumping, could explain these alternating wave attenuation regimes.

Overall, the manuscript is reasonably well written and most conclusions are well supported by evidence. Discussions of limitations and uncertainty are also well incorporated. My main criticism relates to the style of writing, which is (i) quite informal in places and (ii) not efficient, with a lot of repetitions and general lack of conciseness. This makes the paper unnecessarily long in my opinion, which in turn deteriorates the reading experience. Therefore, I strongly suggest that the many authors of this paper have a critical look at the writing and attempt to be more concise in presenting their arguments. This is the main reason why I recommend major revisions. More details and suggestions are provided in my comments below.

We want to thank the reviewer for their detailed comments and helpful suggestions about our work and manuscript. We observe that the reviewer is positive about the observations presented, and that most scientific points raised are relatively minor clarifications that we can easily implement.

Regarding the format and manuscript organization aspects raised by the reviewer, we agree that the reorganization of the manuscript suggested is a good suggestion, and, while this will be quite a bit of work on our end to implement, we are willing to do so. As this is purely a reorganization task, this should in theory not present any fundamental challenge.

We agree that the reviewer suggests good changes that will make the manuscript easier to read and, therefore, we are willing to:

- move some of the more technical / "nitty gritty details" to appendixes
- re-order some of the parts as suggested by the other reviewers, taking some of the points that are for now discussed later in the manuscript earlier on
- cut on redundant parts and / or move some in-depth discussions that can feel redundant to appendixes, as suggested by this reviewer

Naturally, this will be a significant amount of work that will take a bit of time, especially as many contributors have participated in this manuscript.

In the following, we follow The Cryosphere's revision process that, at this stage, only answers to the reviewer are provided (and we do not provide an updated manuscript yet).

Main comments:

- p2: I find the list of references given for the different sources of wave attenuation by sea ice and sea ice breakup to be somewhat biased and missing key papers, especially from key contributors like Squire, Meylan, Bennetts, Montiel, etc. Some suggestions: Mosig et al (2015) for viscoelasticity; Kohout and Meylan (2008), Montiel et al (2016), Pitt and Bennetts (2024) for scattering; Montiel and Squire (2017), Mokus and Montiel (2022) for breakup. In addition, I fail to see the distinction between diffraction and scattering in this context. The paper by Zhao and Shen develops a diffusion approximation from a scattering model in a specific regime and is not really representative of the research on wave scattering in the MIZ.

Thank you for pointing to this. We will take an iteration on the introduction and include the references you suggest, as well as the key papers linked to these. We are fine toning down the distinction between diffraction and scattering.

- I think section 3 is too long and redundant. I understand the authors want to cover their bases, but I think the analysis done in section 3.2 is sufficient to demonstrate that wave-current interactions alone is not enough to explain the observed modulation. Sections 3.1 and 3.3 add very little to the paper in my opinion. My advice would be to focus on the results of section 3.2 and briefly mention that other lines of evidence though ray tracing analysis and altimeter data in open water support the conclusions. Maybe 3.1 and 3.3 could be included as a supplement or appendix if the authors think they are important. In its current form, I don't see the added value of having them in the main text.

We wanted to make sure, as the reviewer points out, to "cover our bases" and analyze this case from several different perspectives to make it as sure as possible that our observations cannot be explained by a "non ice related" mechanism. In particular, our discussions with wave experts, several of whom are part of this paper, made it clear that we had to carefully consider bathymetry and current effects, and whether these could explain the observations (see e.g. discussions about these aspects in https://doi.org/10.5194/egusphere-2024-2104 , https://doi.org/10.1017/jog.2022.99 , https://doi.org/10.3390/jmse12112036 ). To make absolutely sure that, to the best of our knowledge, this cannot be the case, we considered

several approaches, resulting as pointed out by the reviewers to the three relatively heavy sections 3.1, 3.2, 3.3.

However, it seems that both this and the other reviewer are actually convinced enough by the discussion in section 3.2 alone, and both reviewers say that we are maybe overly cautious (and heavy to read) by effectively "triple checking" this result in sections 3.1 and 3.2. Therefore, we are willing to move most of sections 3.1 and 3.3 to Appendixes, and focus this section on a shortened version of section 3.2, moving our extra evidence into Appendixes. This will be an easy change, though it does require a bit of work to rework the flow of the manuscript.

- In section 4, I think the discussion of all physical processes that could explain the observed modulation is not that convincing. Sea ice convergence/divergence will change ice concentration locally, but many processes are likely to damp waves more in tightly pack ice compared to loose ice, including scattering (due to array effects), turbulence and yes also floe-floe interactions. I think this section does not need to be that long, as what it mostly says is that waves are attenuated more in tighter ice packs.

We are willing to slightly cut down on this section, tone down some aspects of the discussion, make it even clearer that we do not have specific evidence for one mechanism versus another, and mention scattering due to array effects (though we are not sure of how well established this is). We agree that we can highlight that the general conclusion is that "we show from field data that damping depends on the level of packing of the ice", and reduce some of the discussions about collisions. However, we still believe that discussing what mechanisms are likely vs. unlikely to produce such an effect is useful, and we want to keep at least part of the discussion about the possible importance of collisions, though we can mitigate it by suggesting that other mechanisms, such as array effects in the context of scattering, could also play a role.

- I think the discussion of the paper is missing an analysis of what is causing convergence/divergence regime shift in the ice drift. I imagine currents and tides, but I don't think the point has been made sufficiently clear. This means that currents and tides are responsible indirectly, i.e. through their effect on the ice, on the observed modulation. If that's indeed the case, why is this not a more common feature observed in other datasets? Have the authors looked at other students showing SWH time series to see if a similar modulation is seen?

Indeed, we believe that currents and tides are the mechanism responsible for the convergence / divergence observed. Experiments with and without tides in the metroms model show a strong influence on sea ice divergence by the tides. Without tides, only undulations of much smaller magnitude, and varying frequency and phase, are present. Clearly, the convergence / divergence is present in both the observations and the models (see Fig. 3). While the model does not offer a causality explanation to the divergence and convergence it produces, we see two possible sources for the convergence and divergence especially in model data: i) tides and currents, or ii) the effect of the wind, either iia) due to shifting wind conditions that apply a stress on the ice and open or close the outer MIZ, or iib) due to the wind triggering inertial oscillations in the water which, when encountering gradients in the sea ice concentration. However, looking at the 2m winds from ERA5 in the

area (see Fig. 1 here), seems to discard this hypothesis. Therefore, we believe that it is most likely that the tide and currents are responsible.

We are not sure why this effect has not been reported in the literature before. One possible explanation is that while it may happen regularly, this effect is seldom so dominant and so clearly visible as it is in our specific case. Given the uncertainties in waves in ice attenuation models, buoy motion, buoy signal noise, etc, it is well possible that less pronounced oscillations in this kind would have been easily overlooked. Similarly, the present event was very easy to spot since it is pseudo periodic over a few days, but a monotonic increase or decrease would be harder to flag and associate to this mechanism. If this hypothesis is correct, then the value of our case is that this effect is so pronounced that it is easy to see and hard to overlook: a "hidden variation" of 20 or 30% of the SWH could have been hidden in other sources of noise and uncertainties, and overlooked in other data. Hopefully, researchers will be on the lookout for such signals in the future.

We actually believe, now that we are aware of this phenomenon, that we see it quite regularly in our data (though maybe not always as pronounced). We can take a few examples from other data that we have released previously, see for example Fig. 2 and Fig. 3. We have been aware of this for a few months now, and we believe that this is quite exciting and deserves further analysis. However, this will require quite a lot of work and time to investigate in details, and we unfortunately do not have this time resource available at the moment. We can mention this and add a few points of discussion in the manuscript, but doing a systematic study across many dataset, while very interesting and a logical next step, goes beyond the scope of our manuscript and would be a large endeavour to be done systematically and robustly.

As a side / anecdotical note, the main author of this manuscript was initially very worried seeing this pattern, as he assumed that this was coming from instrument malfunction - though this worry was quickly dissipated by observing that a clear pattern was visible across instruments of different types, which means this is clearly a real signal. But the only reason why we ended up looking at this event was that it was so obviously visible, even in log scale, in Fig. 2 of the manuscript (which is routinely produced by automated scripts processing OMB data) - otherwise, we would have easily overlooked it. While this is anecdotal, this can explain why this was not reported before, and this can be mentioned in the manuscript.

[Figure]

Fig. 1: ERA5 wind direction and strength at the general location of the BOIs further in the MIZ. We deliberately use a full scale to illustrate the (relative absence of) large scale changes in the wind strength and direction with a 12h period. As visible there, the wind does not have drastic 12h quasi-periodic changes over the time considered that would match the convergence and divergence observed.

Fig 2: 2 more examples of strong periodic SWH modulation. These are only some of the most obvious such modulations involving at least 2 buoys we have found looking at other data after the present work was done, we believe that more events with less pronounced modulation are visible in several dataset. These data are selected from the files available at the following url: https://github.com/jerabaul29/2024_OpenMetBuoy_data_release_MarginalIceZone_SeaIce_OpenOcean/tree/main/Data/2022_AWI_UTOKYO , which come from a deployment in the sea ice North-West of Svalbard in 2022, so a different location and year. The data are openly available on github, and they are described in https://www.nature.com/articles/s41597-024-04281-1 . More "less obvious" events may be present in many other dataset - these are only examples of patterns that are so strong and clear that they are easily caught by the naked eye.

[Figure]

(c)

Fig. 3: SWH plot taken from https://doi.org/10.1016/j.coldregions.2021.103463 , Fig. 16 (c). We believe that this may be another example of such "modulated SWH event happening together with sea ice opening and closing". This is the data from a buoy that was deployed in April 2017 in a field of broken ice floes, around 200km South of the location of the buoys used in our present manuscript. Similarly to our manuscript, strong modulation in the SWH is observed over a period of a few days. There is also strong sea ice opening and closing happening in the area, as revealed by "buoy triangle analysis", see the manuscript for more details. This modulation was not particularly noted back then.

Other comments:

- p3, last sentence: This sentence is hard to read and could be worded better.

We are ready to reformulate this sentence.

- Fig 1: It is not clear which of the buoys on the left panel are selected for the SWH data shown on the right panel. Do the colours of the tracks and curves match? If so, why is there no black dot on the orange track?

Yes, the colors do match. However, we had also included trajectories that were not used as no SWH data was provided by some of the prototype buoys, and the trajectories overlap and hide each others. We will remove these additional trajectories to make the plot easier to read in the next version of the paper.

- p7: "As visible there ..." -> that is not obvious to me. How can we make out the MIZ in this image and where are the BOIs at the time of the image?

The buoy positions are indicated by the markers at the time of the image. We believe that, while it is always challenging to interpret SAR images, the reader should be able to distinguish the general shape of the MIZ limit.

However, we agree that the figure is not so easy to read. Therefore, we are willing to:

- re-generate the figure so that only the BOIs are included, which makes the figure much easier to read (similar to the next point)
- draw the general MIZ limit on top of the figure, to indicate the MIZ limits.

The data are much easier to interact with on the portal, so the user curious for more details should use the link provided. Unfortunately, the portal is not designed for generating and exporting "publication quality figures", so we just have to do the best we can with it. Using another solution would be too much time and effort compared to be implemented.

- Fig 4: It is not clear which tracks correspond to the BOIs. Also I count more than 9 tracks, even though it was mentioned earlier that 9 buoys were deployed.

Indeed, it is not very easy to see which buoys are the BOIs, and there are more trajectories because we also had trackers that only did GPS tracking, not wave measurements. In addition, we had actually deployed more trackers, but there are some buoys that are outside of the area of interest at the moment of the event and, therefore, are not BOIs. The clutter on this figure was due to using the "raw" portal image, and the fact that the portal cannot be tuned to enable or disable individual trajectories in a dataset.

We have worked towards improving this and will now use a tweaked custom version of the portal rendering, see the screenshot below (Fig 3). Unfortunately, the portal is not so flexible for doing the plotting, so this is the best we can do - the users curious of more details can follow the link provided and interactively look at the data if they want.

[Figure]

Fig 4: re-generated portal image, with only the trajectories of interest.

- p8: The discussion of Fig 5 should probably guide the reader towards the conclusion made. It is not clear at all to me that ice floes around the BOIs have similar size to the wavelength. Are dark patches open water? If so, in panel (d) it seems to be the other way around.

We agree that it is often not easy to interpret SAR images, and we believe that it is hard to make more out of these images than just getting an understanding of the general texture and length scales of the floes present in the images. Moreover, it is such, to the best of our understanding, that the distinction between water and ice is not as simple as dark vs. light color, due to variations in the kind of mode used when the images were acquired, polarization, incident angles, etc. What we want to point the reader to is that i) in subfigure (a), the wave crests are clearly visible and correspond well to the wave propagation direction predicted by the model; in subfigures (b) and (c) , it is visible that the SAR return is very inhomogeneous, with typical length scales of O(1km); this likely indicates complex fields of broken floes, and confirms that the area is neither close ice, nor open water, but something "in between" that can support the convergence and divergence obtained from the buoys trajectories and the model, confirming that this is realistically happening. Finally, the image (d) shows that this MIZ where the buoys are located is the outer area at the periphery of a field of larger broken floes. This will be made clearer in the next version of the manuscript.

- Fig 5: annotations on each panel are too small.

It is difficult to really scale up the annotations that are part of each panel, because this is exported directly from the data visualization portal, which provides only limited ability to tune annotations. We will do our best to re-generate these figures and maximize the annotations, but in the end, the reader curious of more details will have to follow the links provided to look

on the portal directly, and we can only do "as good as we can" when exporting figures from the portal into a static manuscript.

- p11: The wave attenuation model used by Yu et al (2022) is not a common choice. Could this choice be better explained? The empirical model was obtained by fitting data obtained in the Southern Ocean Autumn, likely with a lot of pancake ice, so probably very different ice conditions compared to the Arctic Spring.

The Yu et al parameterization is used here as we have observed experimentally in several studies over the last year, both at the Met institute and at MeteoFrance, that this parameterization when used in operational wave models is the one that allows, without ad hoc tuning, to produce wave predictions in the MIZ that statistically best agree with observations. This was reported succinctly in a report (https://documentation.marine.copernicus.eu/QUID/CMEMS-ARC-QUID-002-014.pdf ), and was also discussed recently at the EGU ( https://meetingorganizer.copernicus.org/EGU24/EGU24-18430.html ). This observation that the parameterization of Yu et. al. provides significantly better results than other models available was confirmed independently with both WAM and WW3 runs at MetNo and MeteoFrance, which were run by co-authors on this manuscript.

We have been in contact with J. Yu and E. Rogers over the last months and they were also positively surprised by this result, and at this point we have fully switched to using their parameterization at MetNo in operational models. We believe that the robustness of their parameterization shows empirically that simple models based on tuning to large datasets can be relatively robust independently of the exact location and season, which is an interesting fact. We agree that this may be a bit surprising, and we can make the reasons for this choice clear in an added discussion to the next version of the manuscript.

- Eqs (7), (8), (9): mathematical notations are quite poor in all these equations. Usually successive letters in italic denotes the product of the quantities denoted by the corresponding letters, so for instance $SWH$ actually means S*W*H. Grouping letters together to denote a single quantities is usually done by using roman font type.

Thank you for pointing this out, will we fix this typography and style aspect in the next iteration of the manuscript.

- p27, penultimate sentence: This statement assumes that the ice-induced wave attenuation model used captures properly the dependence on thickness and concentration. I don't think this has been verified.

The waves in ice attenuation parameterization used (Yu et al) has a dependency on sea ice thickness. In addition, the wave model as a whole has a dependency on the sea ice concentration as the terms for the open water vs. fully ice covered contributions in the spectral model are weighted by the sea ice concentration. Therefore, it is, strictly speaking, correct to say that the model takes into account both sea ice thickness and concentration.

However, we agree with the reviewer that this has not been carefully verified yet in the peer reviewed literature (as a side note, we would argue that this is not better or worse than the situation with any other model or parameterization, as we believe that no model has really

been well and robustly verified in general, as visible from the plethora of parameterizations developed and tuned over the years, and the difficulty to make these work robustly in operational models). Therefore, we are willing to tone down this statement, and i) remind the reader of how both effects are taken into account and write that, while ii) making it clear that this is a best attempt at taking these parameters into account, but the approach may still have imperfections.

- p28, second paragraph: I don't understand why refraction, reflection and diffusion are used instead of the more general term "scattering". Also, overwash should be mentioned as a dissipative process, noting that it doesn't fit into the categories listed. Further, scattering does not just depend on floe geometry. There are also array effects (multiple scattering), so the response (including attenuation) will change if floes are loosely or tightly packed, as could be expected in a divergent or convergent ice drift regime, respectively. Therefore, I'm not sure scattering should be dismissed so easily.

We are fine changing our text to use the term scattering, which may indeed be a better technical term in this context, as well as discussing scattering in more detail including the points raised by the reviewer. We are also fine to add a mention about overwash - though it is our feeling, after having been on the sea ice and deploying the instruments, that the freeboard was enough to make overwash relatively unlikely.

- p28, 3rd paragraph: Viscoelasticity is interesting as it has been used to explain attenuation in homogeneous ice cover as well as highly inhomogeneous ice covers. In the latter case, viscoelasticity is of course not the process that causes attenuation, but a convenient effective model for wave damping by sea ice. In the present case, where the ice field seems to be broken up into floes, i.e. non homogeneous, I agree viscoelasticity is likely to not be the dominant physics explaining wave attenuation. I don't think changes in Young's modulus or temperature is the main argument against viscoelasticity.

Thank you for pointing this out. We believe things have been a bit unclear in the literature on this aspect. We are happy that the reviewers agrees that "in the latter case, viscoelasticity is of course not the process that causes attenuation, but a convenient effective model": this is also our view, but we believe that not all papers make it clear, and we are not sure either that everybody in the community would agree (though this is speculations on our end). In particular, it is in our experience quite common to be asked by reviewers to tune or consider tuning "existing models" (that often rely on one form or another of viscoleasticity) to fit observations, even in cases where viscoelasticity is clearly not a realistic mechanism.

We agree with the reviewer's comment, and we will point this out better in the next version of the manuscript: i.e., i) there is no good reason to expect viscoelasticity to play an important role in the present case, and ii) even if it would play some role, there are no temperature fluctuations that could make this vary. We would like to keep the note about the temperature variations, as this also affects / partially rules out the possibility of strong melting / freezing happening back and forth over the period considered.

- p28/29: I think the discussion on collision based on reviews from a different paper is out of place. I'm sorry to hear the authors had a difficult reviewing experience in the past, but I don't think the present manuscript is the place to settle scores. Collision studies have been

conducted, at least in the lab, so why not just refer to those, e.g. Bennetts and Williams (2015).

We thank the reviewers for pointing out that this could be read differently than what we intended. We believe that this comment refers to:

" The question of the existence of collisions in the MIZ has been slightly polemic in the last few years (while no article explicitly mentions this as being polemic, as far as the authors are aware, this assessment is based on personal communications and discussions received by some of the authors, and reviews received during the publication process of Løken et al. (2022)). In particular, there are arguments that floes follow the waves in synchronization, so that there are no collisions actually happening between the floes even if all floes move",

and some of the discussions following this.

Our aim here was not to "settle scores" but rather to explain to the readers that there have been some debate / disagreements in the community, probably to a larger extent than what is possible to grasp from reading the literature (not that disagreement is not bad per se in our opinion; it is our view that constructive disagreement, as long as it is not obstruction of evidences, is much welcome), and that this may have biased the literature a bit - this means there are strong opinions coming from some people who likely regularly act as reviewers, which may effectively filter out the manuscripts that readers go through and modify the relative balance between different phenomena discussed in large parts of the literature.

But we agree that this can be either removed or quite a bit toned down, and we will add the reference suggested and take one more iteration through the literature. Still, we note that the literature discussing these effects is (surprisingly?) scarce, despite the fact that collisions are a reasonable attenuation mechanism that has been measured both in the laboratory and in the field in the past.

---

## Author Response (AR1)

Dear Editor, dear reviewers,

Please find attached our updated manuscript. The point-by-point answer to the reviewers was made available in the discussion comments available at:
https://egusphere.copernicus.org/preprints/2024/egusphere-2024-2619/ .
Our updated manuscript follows the changes discussed there.

Best regards,

Jean Rabault et al.

---

## Author Response (AR2)

**Answer to the Editor**

Dear Editor,

We thank you and the reviewers for your kind words and your help improving our manuscript. We have now addressed the remaining points and we are happy to submit a new revised version of our manuscript, which we hope will be found suitable for publication.

Best regards,

Jean Rabault and co-authors

**Answer to reviewer 1**
**(recommendation: "accepted as is")**

These revisions have indeed made the paper more clear, though I continue to find the process-of-elimination style to be cumbersome for the reader.

Style aside, I continue to think that the main result is a factor of two change in the effective attenuation rate of the waves ice that occurs on a 12 hr cycle. This produces a factor of ten change in the wave heights measured. I understand that the attenuation rates are not well-constrained by the observations, and I did not mean to suggest that the authors attempt to tune their models to reproduce the attenuation specifically. Instead, I was hoping for a reframing that tidal convergence / divergence of sea ice can cause moderate changes in attenuation processes that in turn cause large changes in observed wave height. I suspect that other readers may now draw the same conclusion-- but they will have to work for it.

We want to thank the reviewer for their appreciation of our work. We agree that we can make this conclusion and chain of thoughts even clearer, and this is now added more explicitly into the discussion and conclusion through several additional sentences.

**Answer to reviewer 2**
**(recommendation: accepted subject to technical corrections)**

The authors have made significant changes in response to my comments and I believe the paper is now much improved. I am happy to recommend the paper is accepted for publication after some minor corrections listed below are implemented. I would like to praise the authors for the work they have put in towards the publication of their results.

We want to thank the reviewer for their kind words and their in-depth reviews of our work, which has helped significantly improve the quality of our final manuscript. We have answered the remaining points and updated the manuscript accordingly, see below.

Minor comments (line number refers to track-changes version of the manuscript):

- l87-92: Please rephrase these sentences so they don't start with "See ..."

Done.

- Fig 6: Having individual captions for each panel is a little weird. Why not have a single caption for the whole figure, referring to each panel, which is more standard. I'm just pointing this out and will let the journal editor/publisher decide whether this is appropriate.

Thank you for your comment. We believe this is largely a question of taste, and we like quite well the split between "detailed subcaptions" under each panel, and "high level caption" below all. This is, in our opinion, more convenient when permalinks are provided. So, we plan on keeping this as is if not there is an issue with the journals template.

- l285: I'm not sure "waves-in-ice interaction mechanisms qualifies as "well-known physics".

We are fine removing "well-known physics", done.

- l293: the report can be included in the ref list I imagine. Also please rephrase "reported in a report"!

The report is now a reference instead of a footnote, and we have fixed the formulation.

- Eq (5): SWH missing in the first term on the numerator.

Thank you for pointing this typo, this is now fixed.

- Sections 3.2 and 3.3 are now very short. I suggest removing them and incorporating the corresponding statements in section 3.1.

We agree that these sections are now very short since, as suggested by the authors, their contents have been moved to Appendixes. However, we believe that they do not belong well to section 3.1 from a "logics" perspective, so we would like to keep them separate as is now.

- l609: The authors make excessive use of "per se". Most of the time, this is not needed, so I suggest removing.

Thank you for pointing this out. We have removed some of the "per se" and kept some others, depending on how much we want to keep the emphasis on the points we make.

- l720: I suggest using \Delta x, instead of dx, as dx is a differential which has a specific mathematical meaning.

Fixed, and same for \delta t where we changed to \Delta t to be consistent.

- l974-975: what's the difference between closedness and compactness? Both of them are used throughout the manuscript. I suggest sticking with one for consistency.

We agree that there is no good reason to use two distinct words - we have changed to using "closedness" everywhere.

---

## Author Response (AR3)

Dear Editor,

Thank you for your support and the productive review process of our manuscript.

I have now modified Fig. 6 as you had requested, so that there are no subfigures any longer.

Best regards,

Jean Rabault et. al.